# Epigenetic specifications of host chromosome docking sites for latent Epstein-Barr virus

Kyoung-Dong Kim[1], Hideki Tanizawa [2], Alessandra De Leo[3], Olga Vladimirova[3], Andrew Kossenkov[3], Fang Lu[3], Louise C. Showe[3], Ken-ichi Noma [2] & Paul M. Lieberman[3]*

Epstein-Barr virus (EBV) genomes persist in latently infected cells as extrachromosomal episomes that attach to host chromosomes through the tethering functions of EBNA1, a viral encoded sequence-specific DNA binding protein. Here we employ circular chromosome conformation capture (4C) analysis to identify genome-wide associations between EBV episomes and host chromosomes. We find that EBV episomes in Burkitt's lymphoma cells preferentially associate with cellular genomic sites containing EBNA1 binding sites enriched with B-cell factors EBF1 and RBP-jK, the repressive histone mark H3K9me3, and AT-rich flanking sequence. These attachment sites correspond to transcriptionally silenced genes with GO enrichment for neuronal function and protein kinase A pathways. Depletion of EBNA1 leads to a transcriptional de-repression of silenced genes and reduction in H3K9me3. EBV attachment sites in lymphoblastoid cells with different latency type show different correlations, suggesting that host chromosome attachment sites are functionally linked to latency type gene expression programs.

[1] Department of Systems Biotechnology, Chung-Ang University, Anseong, Korea. [2] Institute of Molecular Biology, University of Oregon, Eugene, OR 97403, USA. [3] The Wistar Institute, 3601 Spruce Street, Philadelphia, PA 19146, USA. *email: lieberman@wistar.org

Epstein-Barr Virus (EBV) is a ubiquitous human gamma-herpesvirus that has been etiologically linked to various cancers, including Burkitt's lymphoma, Hodgkin's disease, nasopharyngeal carcinoma, gastric carcinoma, and post-transplant lymphoproliferative disease[1,2]. EBV-associated tumors harbor latent forms of EBV genomes that express a restricted set of viral genes, many of which have been implicated in tumorigenesis[3,4]. The different patterns of latent viral gene expression are referred to as EBV latency types and are known to depend on epigenetic factors related to the host cell or tumor origin[5,6]. However, our knowledge of the epigenetic mechanisms that determine these different patterns of viral gene expression remain incomplete.

Chromatin structure and epigenetic modifications contribute significantly to the regulation of EBV latency[5,7]. In the majority of EBV latent infections, the viral genome persists in the nucleus as a chromatinized, covalently closed, circular genome that closely tracks with the host chromosome during cell division cycle. Several latency-associated proteins contribute to the epigenetic control of both viral and host gene expression[8,9]. EBNA2, EBNA-3s, and EBNA-LP function as transcriptional co-activators and co-repressors in conjunction with cellular sequence specific factors, such as RBP-jK and EBF1[10–12]. In contrast, EBNA1 is a sequence specific DNA binding protein required for viral DNA replication and episome segregation during cell division[13,14]. The EBNA1 DNA-binding domain (DBD) has structural similarity to the DBD of KSHV LANA and HPV E2[15–17]. EBNA1 DBD can bind with sequence specificity to the viral genome at consensus binding sites within OriP and at the EBNA1 promoter, termed Qp[18]. In addition, EBNA1 can bind with high affinity to ~1000 sites in the host chromosome, several of which have demonstrated functions in transcriptional regulation of host genes[19,20]. However, the majority of the EBNA1 binding sites in the host chromosome have unknown function.

Viral episome segregation during cell division depends on the metaphase chromosome attachment function of EBNA1[21]. The metaphase chromosome binding domains of EBNA1 are separate from the C-terminal DBD and have been mapped to amino acids 8–54, 72–84, and 328–365[22–26]. These domains are enriched in arginine-glycine (RG) and have been shown to possess AT-hook DNA binding[27,28], as well as G-quadruplex RNA binding[29,30], and to interact with the host RNA-binding protein EBP2[26,31], as well as other nuclear factors, such as HMGB2[22], nucleolin[32], and RCC1[33]. Adjacent to the RG-motifs is a zinc-hook motif that is capable of Zn mediated homotypic interactions[34]. These regions of EBNA1 have also been referred to as the linking domains as they confer the ability of EBNA1 to self-aggregate in biochemical assays[35–39]. Precisely how EBNA1 metaphase attachment occurs and whether this is limited to metaphase chromosomes or occurs similarly during interphase is not yet known. Nor is it known whether the EBV genome is tethered to certain specific sites in the host chromosome, and whether these sites are determined by any genetic or epigenetic signatures.

Chromosome conformation capture (3C) methods have been devised for measuring DNA conformation and loop interactions[40]. 3C has been used to study DNA loop interactions in the EBV genome and how these can vary between latency types in accord with promoter activity[5,41,42]. In these studies, OriP was identified as a central hub for the EBV genome and its interactions varied with the activity of different promoters utilized in different latency types. Here, we utilize circular chromosome conformation capture (4C) to identify attachment sites in the host chromosome where the EBV genome tends to localize[43–45]. In latently infected BL cells, we find that EBV is frequently associated with host chromosome sites enriched for sequence-specific EBNA1 binding sites, along with binding sites for EBF1 and RBP-jK, and enrichment of H3K9me3 and AT-rich DNA. We propose that EBV tethering occurs, in part, through EBNA1 homotypic interactions. We also find that EBV host chromosome attachment sites can depend on the host-cell and latency type.

## Results

**Detection of EBV tethering sites on human chromosomes.** To begin to understand how EBV episomes interact with host chromosomes, we applied 4C assay to two EBV positive Burkitt's lymphoma cell lines, MutuI and Raji. These cell lines maintain a relatively high-copy number of viral episomes and express a restricted type I latency over many generations[46]. The 4C method was selected to obtain higher resolution than Hi-C and other related methods. To achieve high resolution, chromatin was digested with 4 bp cutter MboI and ligated in the nucleus followed by in situ Hi-C protocol[47]. Subsequently, DNA was extracted and subjected to a second round of digestion with the 4 bp cutter Csp6I. The digested DNA was re-ligated to form circular second ligation product which served as template for library generation with specific primers originated from the EBV genome to capture cellular DNA associated with EBV episomal DNA (Fig. 1a). To generate the 4C library, we chose several different bait positions on the EBV genome including the EBNA1-bound elements at the FR, DS/Cp, and Qp regions, and one CTCF-cohesin binding site at the LMP1/2 locus. We also designed two different primers, DS/Cp-1 and DS/Cp-2, from the same fragment in DS/Cp region to check the reproducibility of 4C libraries (Fig. 1b and Supplementary Table 1). In addition, we prepared two biological replicates for all samples. Sequencing reads were processed and EBV-associated 4C peaks on the host chromosome were generated (Supplementary Fig. 1a and Supplementary Table 2, 3). We identified numerous EBV 4C peaks throughout chromosomes with 10 kb resolution indicating that EBV tethering sites can be identified at the gene level and at higher resolution than previous studies[48]. The overall pattern of EBV 4C peaks from 5 baits revealed common peaks at specific genomic regions and a general distribution of peaks throughout all chromosomes. We found highest similarity of peaks among primers FR-L (Left side), DS/Cp-1, and DS/Cp-2, which surround the OriP region of the EBV genome (Fig. 1c and Supplementary Fig. 1b). We examined the reproducibility of biological replicates. The Pearson's correlation coefficient was as high as 0.95 for DS/Cp-1 or 2 (Supplementary Fig. 1c, d). We also examined the reproducibility of datasets from two sides of a bait, MboI or Csp6I. The Pearson's correlation coefficient was 0.66–0.77 for FR-L, 0.81–0.93 for DS/Cp-1, and 0.55–0.70 for LMP1/2 (Supplementary Fig. 1c, e). And we examined the similarity of EBV 4C peaks between MutuI and Raji cells and observed 0.65–0.91 of high correlation coefficient from FR-L and DS/Cp-1,2 samples, respectively (Supplementary Fig. 1c, f). Overall, there was strong correlation coefficient of biological replicates and strains for bait primers at the FR-L and DS/Cp-1,2, with lesser correlation at Qp and LMP1/2 regions. This increase correlation may be due to the contribution of OriP, oriented between FR-L and DS/Cp-1,2 in mediating the stronger and more direct interactions between EBV and the host genome. The reproducibility of datasets from the OriP bait regions suggests that these correspond to EBV tethering sites on host chromosomes.

**Significant EBV tethering sites over the human chromosomes.** To identify EBV tethering sites on the human chromosomes, we focused on 4C peaks of the most highly correlated datasets from DS/Cp-1 and DS/Cp-2 (Fig. 1c). We defined common 4C peaks as those that occurred as significant in at least 5 out of 6 total experimental datasets (Supplementary Fig. 2a, b). To exclude the

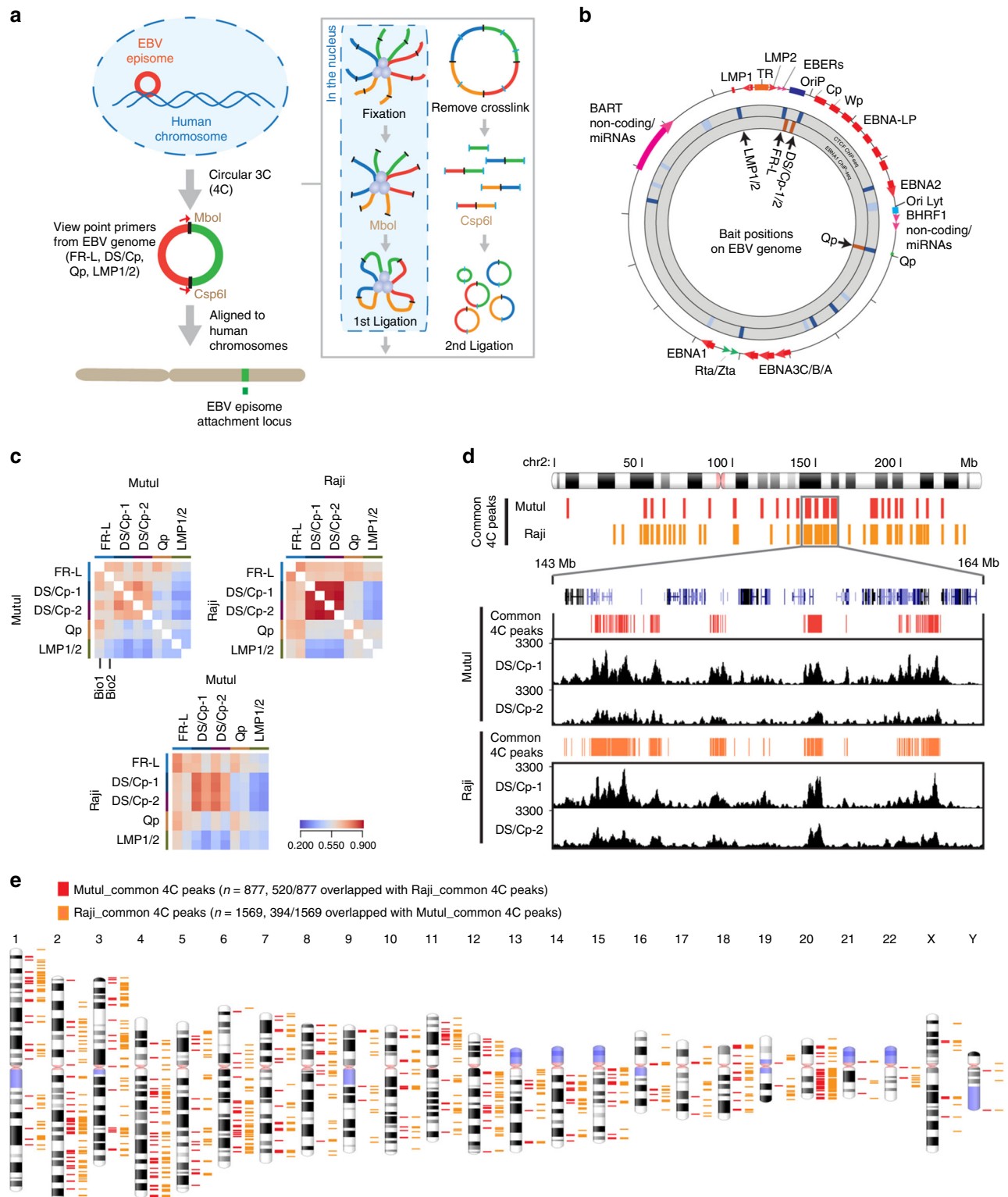

**Fig. 1 4C analysis of EBV-host DNA interactome. a** The 4C experimental procedure. **b** Baits positions at FR-L (Left), DS/Cp, Qp, and LMP1/2, on EBV genome. The positions of EBV genes and binding sites for CTCF and EBNA1 are indicated. **c** Heatmap showing the Pearson's correlation coefficients of 4C-seq scores for 10 kb non-overlapped windows between baits from same (top) or different strains (bottom). **d** Example of 4C peaks for Mutul (DS/Cp-1,2) and Raji (DS/Cp-1,2) at the chr2: 143–164 Mb region with 10 kb resolution. **e** The 877 common 4C peaks for Mutul (red bars) and 1569 common 4C peaks for Raji (orange bars) are loaded on human all chromosomes with 10 kb resolution.

reads that may have originated from a rare EBV integration event on the host genome, we compared our common 4C peaks for Raji cells with the database of viral integration sites previously reported in Raji cells[49]. We found the number of overlap sites between two datasets to be negligible since only 8 out of 1561 common 4C peaks had any overlap, suggesting that our 4C dataset is acquired primarily from the association between non-integrated EBV episomes and host chromosomes (Supplementary Fig. 2c). We identified 877 and 1569 common 4C peaks for MutuI and Raji, respectively. There was substantial overlap of peaks between these two cell types, with 520/877 common 4C peaks for MutuI overlapping with Raji and 394/1569 common 4C peaks of Raji overlapping with MutuI (Fig. 1d). These peaks were distributed across all chromosomes (Fig. 1e). The chromosome distribution was also shown by Circos diagrams (Supplementary Fig. 2d, e) and revealed that EBV tethering sites are located throughout all chromosomes, with some enrichments on chromosomes 2, 4, 6, 12, and 18, and some under-representation of chromosomes 9, 19, 21, and 22 (Fig. 1e and Supplementary Fig. 2d, e).

**Validation of 4C with FISH**. We used fluorescence in situ hybridization (FISH) to validate several of the 4C common peaks observed in both Raji and MutuI. First, we counted the average number of EBV FISH signals on several host chromosomes. The target chromosomes were recognized by cellular FISH probes which hybridize to ~100 kb region of chromosome 2, 4 12, and 17 (Supplementary Table 4). We observed that chromosome 2 and 12 have many more EBV FISH signals than chromosome 17 (Supplementary Fig. 3a, b), which is consistent with 4C data. To validate attachments at specific chromosome loci, we generated FISH probes from bacmids situated at 4C common peaks for chromosome 2q24.1, 2p16.3, 2q24.2, 4q34.2, and 12q21.2 and negative control sites at 2p24.2 and 17pter (Fig. 2a, b). We first demonstrated that 4C common sites at 2q24.1 colocalize with EBV FISH signal at higher frequency in multiple interphase MutuI cells (Fig. 2c). We further tested co-localization between EBV and genomic loci in interphase MutuI cells and revealed that 4C targets overlapped with EBV genome at a rate of 30–40% while negative control (n.c.) regions colocalized at less than 10% frequency (Fig. 2d, e and Supplementary Fig. 3c). Similar results were obtained with Raji cells (Fig. 2f and Supplementary Fig. 3d).

A major question is whether EBV associations with cellular chromosomes are fundamentally different for condensed mitotic chromosome in comparison to interphase chromosomes. Therefore, we examined the co-localization between EBV and genomic loci in MutuI mitotic cells. We found that 4C interaction sites are overlapped with EBV at a frequency of 40–50% at 4C common sites, while the negative control sites overlapped with a frequency below 10% (Fig. 2g, h and Supplementary Fig. 3e). This indicates that 4C interaction sites in interphase cells correlate well with attachment sites on mitotic chromosomes measured by FISH for both MutuI and Raji cells. These results collectively suggest that attachment of EBV episomes to the host human genome is maintained during interphase and mitosis.

**Epigenetic features of EBV tethering sites**. To identify chromosomal proteins and epigenetic features that may specify EBV tethering, we examined the correlation of EBV 4C data with other ChIP-seq data, as well as AT richness and gene density (Fig. 3). We checked the correlation of EBV with EBNA1, EBF1, RBP-jK, CTCF, and histone markers, H3K9me3, H3K4me3, and H3K27ac in MutuI cells by heatmap comparison and average enrichments (Fig. 3a–c and Supplementary Table 5). The heatmap comparison

showed that EBNA1, EBF1, RBP-jK, H3K9me3, and AT rich sequence have a significant correlation with the enrichment of EBV 4C peaks (Fig. 3a). The heatmap enrichment was divided into 10 groups and assessed by line graphs to further demonstrate the correlation of EBV 4C peaks with EBNA1, H3K9me3, and AT rich sequence (Fig. 3b). Interestingly, the ratio of AT sequences is enriched at regions flanking EBV 4C peaks, but it is a lower at the 2–3 kb center of EBV peaks. Furthermore, the average enrichments of EBNA1 and H3K9me3 at EBV 4C target site were significantly higher than at EBV non-4C target sites. EBF1 and RBP-jK have overlaps at EBV 4C sites judged by box graph for average enrichment (Fig. 3c), but these enrichments were not accelerated at the strongest 4C targets (Fig. 3a, b), implying that EBF1 and RBP-jK have stronger binding sites independent of EBV 4C peaks. The peak accumulations of EBNA1, EBF1, RBP-jK, and H3K9me3 were also observed in the genome browser views (Fig. 3d). The EBNA2 and EBNA3 in MutuIII, PolII and H3K27ac in GM12878, and gene density are not correlated with the enrichment of MutuI 4C peaks (Fig. 3a, c). 4C peaks containing EBNA1 binding sites were further enriched for RBP-jK, H3K9me3, and EBF1 (Supplementary Fig. 4a). Inclusion of 4C datasets from the FR region also showed strong co-localization with cellular binding sites for EBNA1 (Supplementary Fig. 4b). These correlation data suggest that EBV tethering favors chromosome locations containing cellular binding sites for EBNA1 with correlated enrichments for EBF1, RBP-jK, H3K9me3, and AT-rich flanking sequences.

**Difference of EBV tethering sites according to latency type**. We examined by computational methods the EBV-host chromosome association for published Hi-C data sets from the EBV positive LCL GM12878[47]. The EBV contacts with the host chromosomes were extracted from the complete host genome interaction network in Hi-C data. We also attempted 4C analysis of an LCL (Mutu-LCL) cells, but due to the low read count, our results did not achieve the same statistical robustness as that for MutuI and Raji (Supplementary Fig. 5a). The low read count in Mutu-LCL may have been due to fewer episome copy numbers, although quantification of viral genomes by qPCR showed similar levels of total EBV DNA in all three cell lines (Supplementary Fig. 5d). Nevertheless, we defined significant peaks for two Mutu-LCL samples by scoring each 10 kb window with the value of surrounding 500 kb. We found significant correlation between LCL 4C and GM12878 Hi-C as shown by heatmap comparison and line graphs (Supplementary Fig. 5b, c). We identified 331 and 636 significant peaks for two Mutu-LCL samples and revealed that overall peak distributions of Mutu-LCL were correlated with GM12878 Hi-C (Supplementary Fig. 5e, f), suggesting that 4C assay is a preferred method to capture EBV attachment sites probably due to its specificity to extract EBV-host genome contacts. We also found that 4C in LCL showed better correlation with Hi-C in LCL GM12878 than with 4C in MutuI or Raji (Supplementary Fig. 5g).

To better understand the potential differences in EBV tethering sites found in LCLs, we compared the common 4C peaks from MutuI and Raji with the interaction sites determined for LCL GM12878. For this analysis, we relied on the Hi-C data for LCL GM12878, as it provided greater resolution and depth than our 4C analysis in LCL. Analysis of the Hi-C in GM12878 did not show a strong correlation of EBV-host interactions with EBNA1 or any of the other features found for MutuI and Raji. In contrast to our comparison between MutuI and Raji 4C, the Hi-C data from LCL GM12878 was not well-correlated with common 4C peaks from MutuI and Raji cells by heatmap comparison and average enrichments (Fig. 4a, b). Only 24 or 54/568 Hi-C peaks

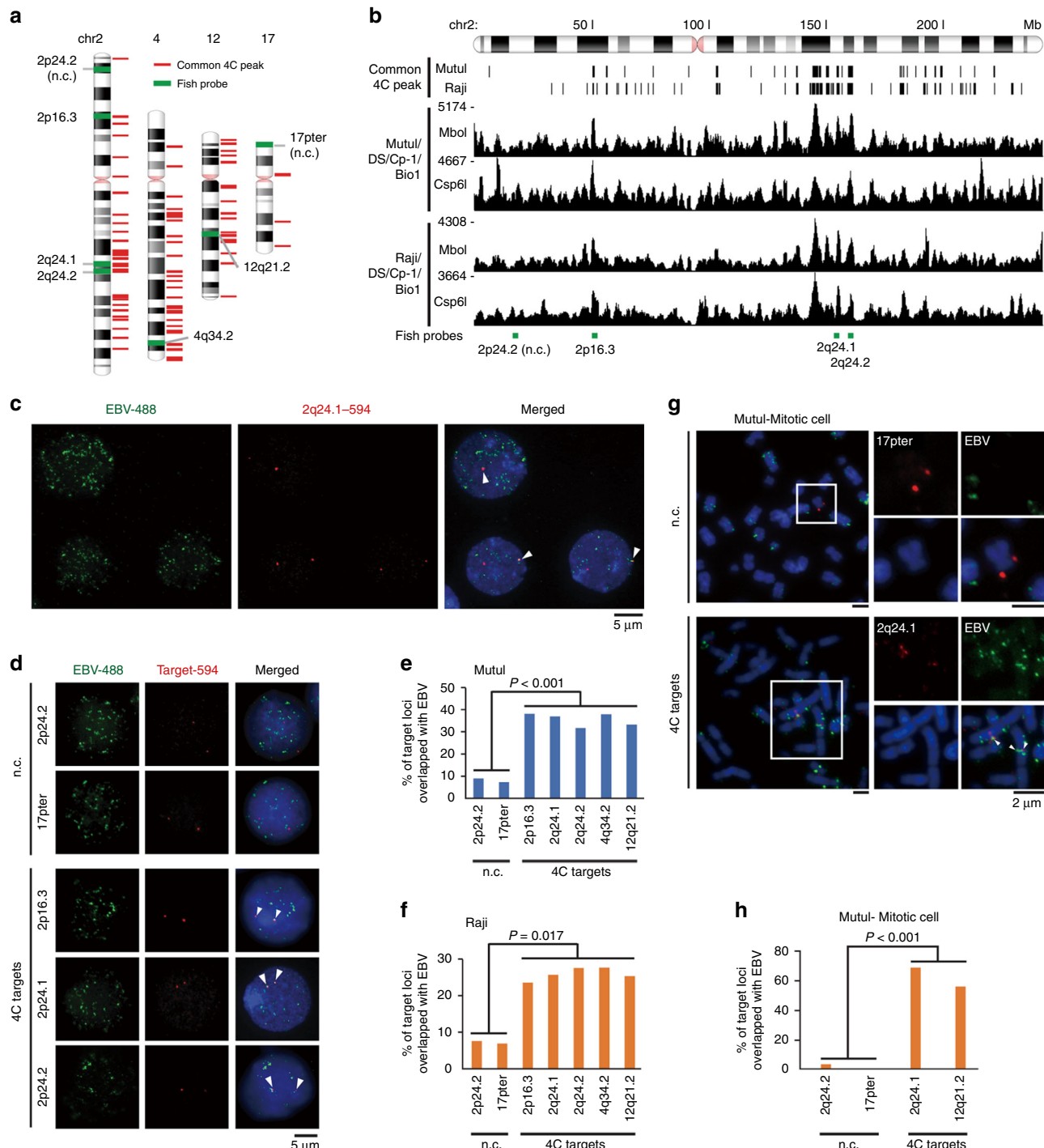

**Fig. 2 Validation of 4C interaction sites by FISH. a** Positions of FISH probes (green bars) on chr2, 4, 12, and 17. **b** Precise positions of FISH probes (green bars) on chr2 with common 4C peaks. **c** Representative images for interphase MutuI nuclei stained with dual FISH for EBV-488 (Oregon Green 488) and Target-594 (Alexa Fluor 594). **d** Representative images of interphase MutuI nucleus stained with dual FISH for EBV-488 and Target-594. **e, f** The percentage of co-localized FISH signals between EBV-488 and Target-594 in interphase nuclei from MutuI (panel **e**) and Raji (panel **f**) cells. The FISH signals were measured in more than 100 nuclei. **g** Representative images of mitotic MutuI nuclei stained with dual FISH for EBV-488 and Target-594. All arrows indicate co-localized FISH signals between EBV-488 and Target-594. **h** The percentage of co-localized FISH signals between EBV-488 and Target-594 in mitotic chromosomes from MutuI cells. The FISH signals were measured in more than 50 nuclei. Error bar represents the standard deviation and P-value were calculated by two-sided Student t-test.

for GM12878 are overlapped with MutuI or Raji, and overall peak distributions of Hi-C peaks for GM12878 are also not correlated with 4C peaks from MutuI and Raji cells (Fig. 4c, d). To obtain a global genomic view of where EBV contact on human chromosome, we categorized EBV tethering sites by type of genetic elements. We found that in MutuI or Raji cells EBV associates more with intergenic regions and less associates with promoters, while in LCL GM12878 cells EBV is more enriched at gene promoters (Fig. 4e). We also observed that while heterochromatic mark H3K9me3 is enriched at the EBV tethering sites in MutuI

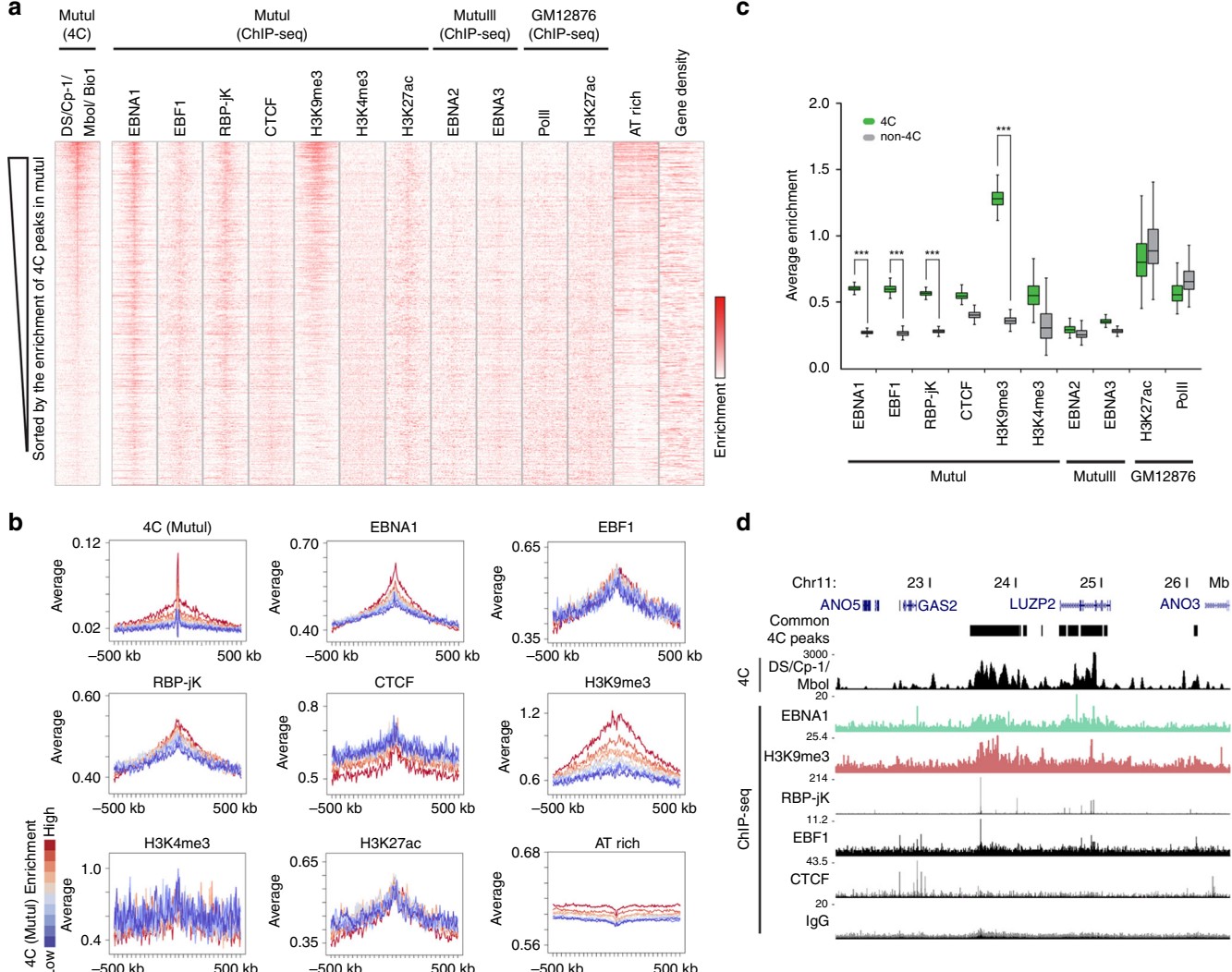

**Fig. 3 Epigenetic signature of EBV tethering sites. a** Heatmap comparison of 4C (MutuI cell, DS/Cp-1 bait, MboI restriction site) with other ChIP-seq data, AT rich sequence and gene density (5 kb bin, 500 kb range). ChIP-seq scores were converted to log values. Red color corresponds to maximum score. White color corresponds to average score. **b** Average sequence enrichment for the indicated ChIP-seq proteins and AT rich sequence classified into 10 groups based on the 4C (MutuI) scores (top: red → bottom: blue). **c** Box plot quantitation of ChIP-seq intensity overlapping 4C peaks or non-4C peaks. Boxplots show center quartiles, midlines show medians and whiskers extend to the data points, which are no more than 1.5× the interquartile range from the box. Error bar represents the standard deviation and *P*-value were calculated by two-sided Mann–Whitney *U* test. **d** UCSC genome browser image showing the comparison of 4C with other ChIP-seq data along the indicated chromosome.

and Raji, more active histone markers such as H3K27ac, H3K4me1, and H3K4me3, along with PolII and EBNA2 are enriched at the EBV tethering sites in GM12878 (Supplementary Fig. 6a–c). These findings suggest that EBV tethering sites on host chromosomes are substantially different between MutuI/Raji BL cells and GM12878 LCL cells.

**Strong EBV tethering sites are enriched in neuronal genes.** Since our 4C datasets in BL cells can reach to a few kilobase resolution, we could identify genes that were in close proximity to 4C common peaks. The Ingenuity Pathway Analysis (IPA) of the genes positioned in common 4C regions revealed that EBV 4C datasets from both MutuI and Raji are significantly associated with genes involved in the protein kinase A signaling, Wnt/Ca$^+$ signaling, and melatonin signaling (Supplementary Fig. 7 and Supplementary Table 6). Previous studies have suggested that EBV modulates the Wnt pathway by the repression of E-cadherin[50], but it is not otherwise obvious how these 4C

interactions correspond to cellular gene regulation. To get more significant interaction targets, we focused on the top 10% of 4C peaks (based on width and read count, Fig. 5a) and found 70 sites that overlapped with these peaks in both Raji and MutuI cells (Fig. 5a, b). We demonstrated that these 70 regions are positively correlated with the average and maximum strength of reads in both cell lines (Fig. 5c) and broadly distributed across multiple chromosomes (Fig. 5d). Gene ontogeny analysis indicated that many of the EBV-tethering site associated genes have functions related to brain and nerve (Fig. 5e and Supplementary Table 7). We show specific examples for EBV interactions at NRXN1, NAV3, and SYT1 which are specifically expressed in brain (Fig. 5f, g). These findings suggest that EBV tethers to gene loci that are likely to be silenced in B-lymphocytes.

**Genes in EBV tethering sites are transcriptionally repressed.** To investigate the potential functional significance of EBV tethering with specific genes, we examined the transcriptional regulation of

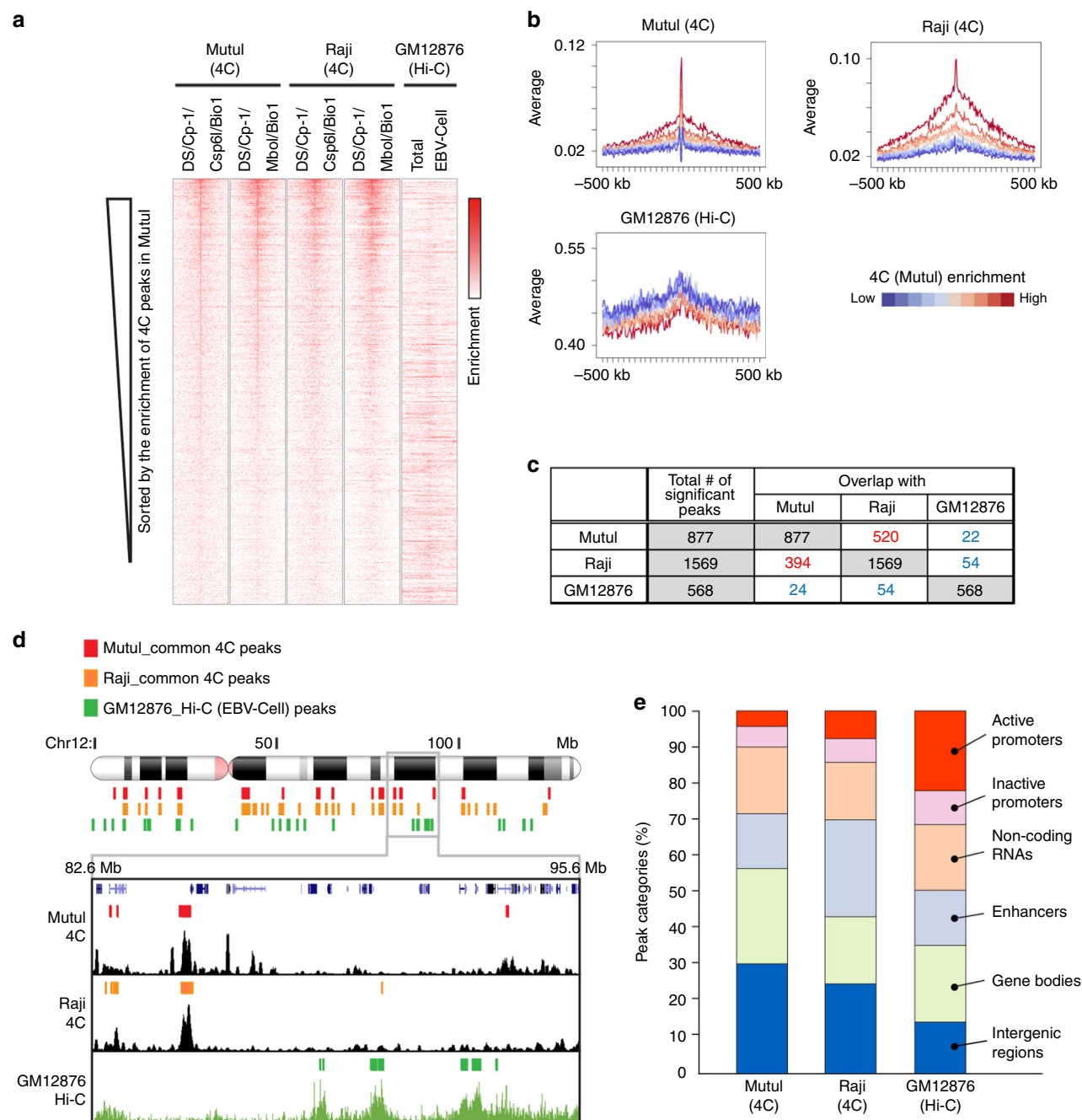

**Fig. 4 Differences between LCL and BL cell tethering sites. a** Heatmap comparison of 4C in MutuI and Raji (DS/Cp-1/Csp6I/MboI) with Hi-C in GM12878. **b** Average enrichment of 4C peaks (DS/Cp-1/Csp6I) in MutuI and Raji and Hi-C peaks in GM12878 shown in panel **a**. The average enrichments are classified into 10 groups based on the 4C (MutuI/DS/Cp-1/Csp6I) scores (top: red → bottom: blue). **c** Table showing the number of overlapped peaks among MutuI (4C), Raji (4C), and GM12878 (Hi-C). **d** Example of 4C peaks for MutuI (DS/Cp-1/Csp6I/Bio1), Raji (DS/Cp-1/Csp6I/Bio1), and Hi-C peaks for GM12878 at the chr12: 82.6–95.6 Mb regions. **e** Distribution of EBV associating genomic loci at the indicated genetic elements.

**EBV 4C-associating genes.** Since the highest frequency EBV tethering site overlapped with EBNA1 binding sites, we first assayed the transcriptional expression of 4C target genes in Raji cells after shRNA depletion of EBNA1 (Fig. 6). EBNA1 was partially depleted by lentivirus transduced shRNA in Raji cells as measured by Western blot (Fig. 6a) and by ChIP assay (Supplemental Fig. 8a, b). As expected, EBNA1 depletion led to a global reduction in EBV genome copy number as measured by FISH (Supplementary Fig. 8c, d). EBNA1 depletion also led to a loss of EBNA1-host chromosome tethering as measured by 3C analysis

of a specific high-frequency 4C peak at chr2:15.6 Mb (Supplementary Fig. 8e, f). RNA analysis of several 4C associated genes revealed a significant upregulation after EBNA1 depletion (Fig. 6b, c). To determine if this trend held genome-wide, we analyzed the average expression of Top10% 4C target genes compared to other genes. This analysis revealed an average ($p <$ 0.01) lower expression level for Top10% 4C target genes compared to non-4C target genes in MutuI and Raji cells (Fig. 6d, e). This trend was further corroborated by comparing the average expression of 4C target genes to non-4C target genes (Fig. 6f). 4C

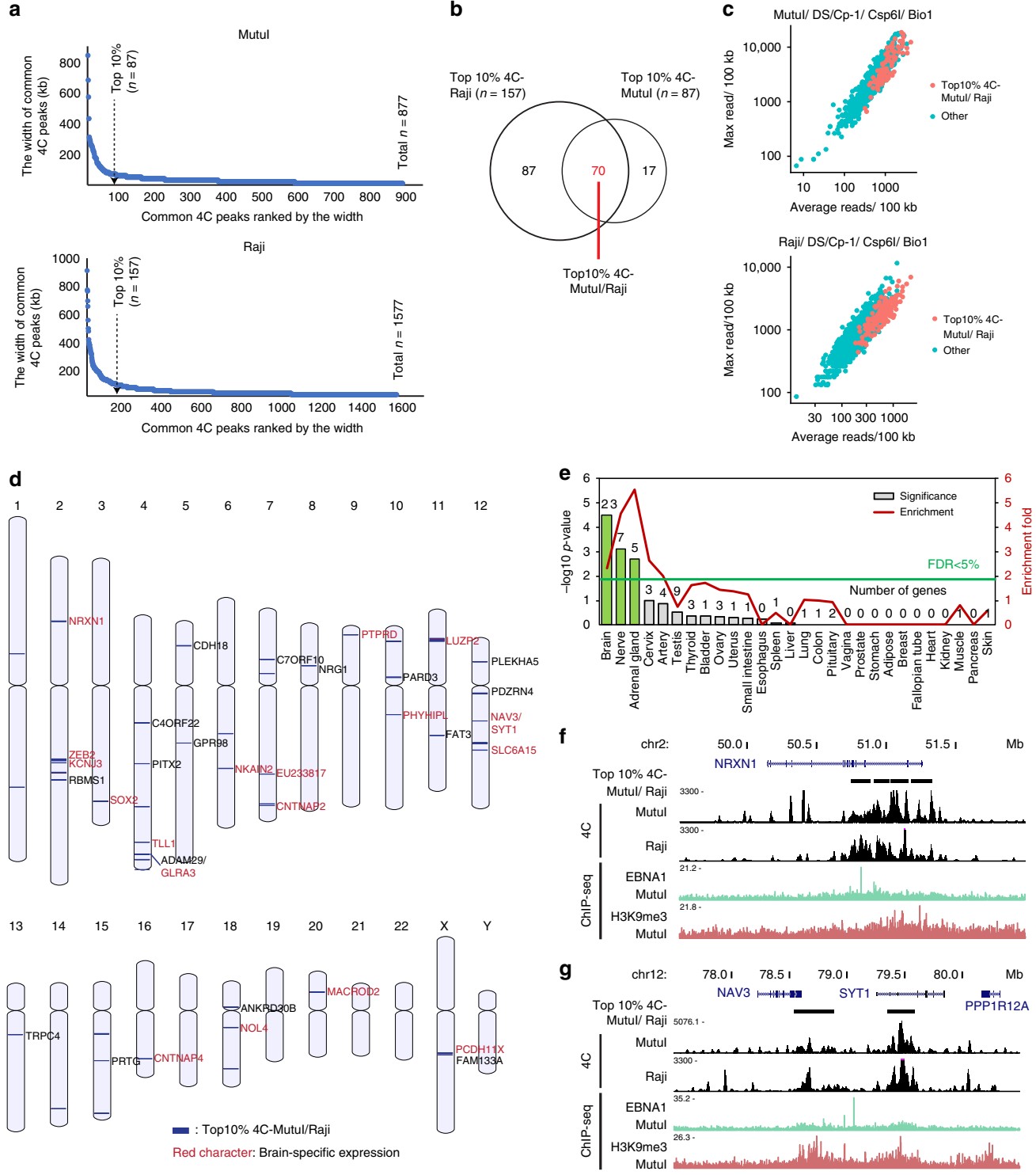

**Fig. 5 High-frequency EBV tethering at transcriptionally silent neuronal genes. a** Common 4C peaks ranked by the width of each peak for MutuI (top panel) and Raji (bottom panel) cells. **b** Venn diagram showing co-occupancy of top 10% common 4C peaks between MutuI and Raji (Top10% 4C-MutuI/ Raji). **c** Distribution of Top10% 4C-MutuI/Raji on the graph of average reads per 100 kb (x-axis) and the maximum read per 100 kb (y-axis) acquired from MutuI/DS/Cp-1/Csp6I/Bio1 (top panel) and Raji/DS/Cp-1/Csp6I/Bio1 (bottom panel). **d** Positions and associated genes for Top10% 4C-MutuI/Raji across chromosomes plotted using NCBI genome decoration page. **e** Enrichment of tissue specific genes positioned in Top10% 4C- MutuI/Raji sites. **f**, **g** UCSC genome browser image showing the comparison of 4C data for MutuI and Raji with EBNA1-/H3K9me3- ChIP-seq data for MutuI along the indicated chromosome regions containing genes NRXN1 (panel **f**), NAV3, and SYT1 (panel **g**) specifically expressed in brain.

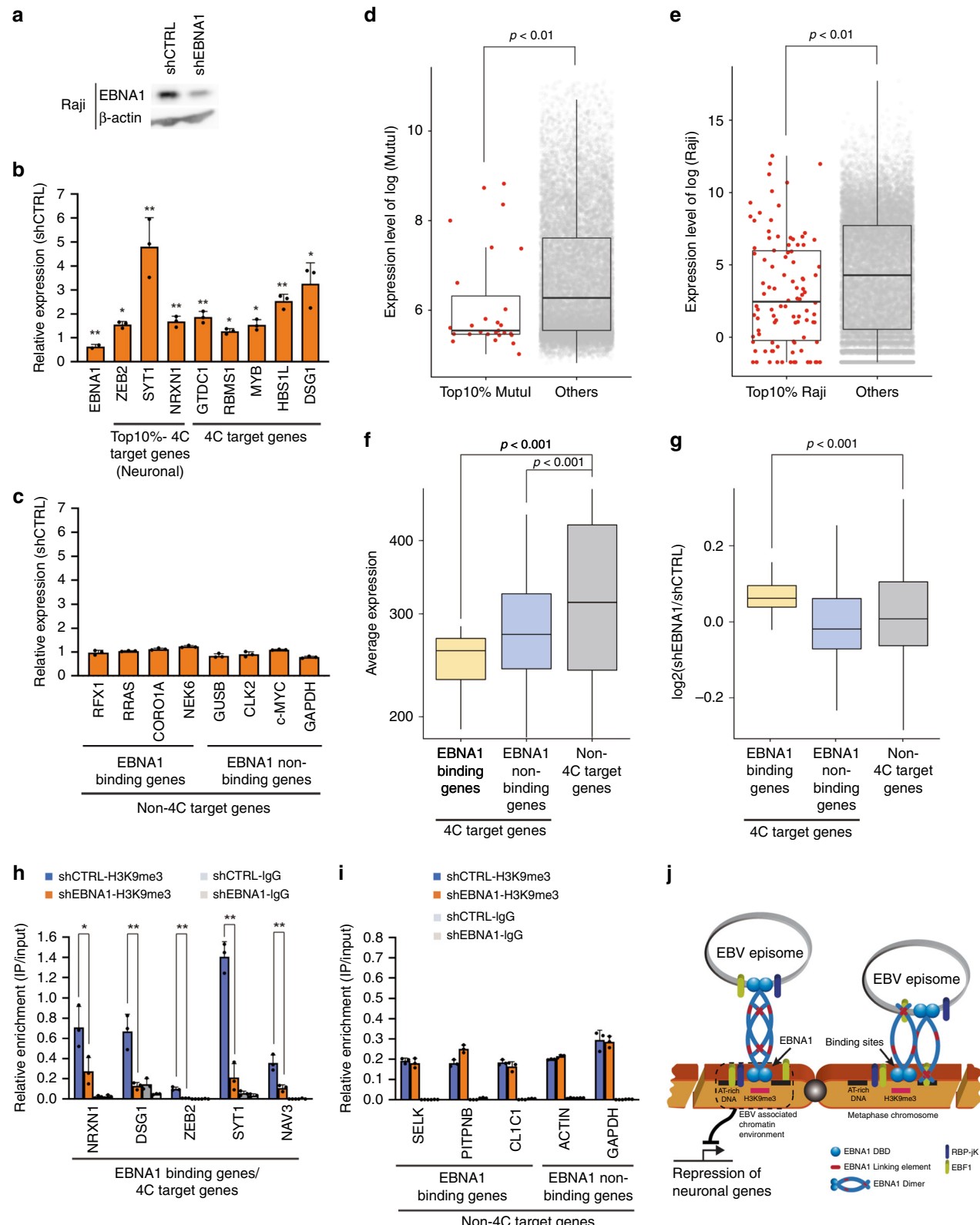

genes with EBNA1 binding sites showed a significantly lower ($p < 0.001$) average expression than non-EBNA1 bound and non-4C genes (Fig. 6f). Furthermore, shEBNA1 depletion followed by RNA-seq analysis revealed a significantly greater ($p < 0.001$) increase in transcription for EBNA1 bound 4C genes relative to non-EBNA1 bound and non-4C target genes (Fig. 6g). shEBNA1 also led to a loss of H3K9me3 at multiple EBNA1 bound 4C-

target genes (Fig. 6h), but not at EBNA1 bound non-4C genes, nor at non-EBNA1 non-4C genes (Fig. 6i). These findings indicate that most cellular genes located at EBV tethering sites are transcriptionally repressed, and that EBNA1 binding at these sites, which is essential for EBV tethering, contributes to their repression mediated, in part, by H3K9me3 associated hetero-chromatin (Fig. 6j).

**Fig. 6 EBNA1-dependent transcriptional repression of host genes at EBV docking sites. a** Raji cells were transduced with lentivirus containing shCTRL (Control) or shEBNA1 targeting vectors. Cells were selected for puromycin resistance and assayed at 7 days post-selection by western blotting for EBNA1 or Actin. **b**, **c** RT-qPCR analysis of mRNA expression in Raji cells transduced with shCTRL or shEBNA1 for EBNA1 or cellular top 10%-4C neuronal target genes or general 4C target genes (panel **b**) or non-4C target genes (panel **c**). RNA expression is shown as fold over shControl and p-values are calculated relative to shControl. Error bar represents the standard deviation and P-value were calculated by two-sided Student t-test. n = 3. **d**, **e** The average mRNA expression for genes positioned in Top10% 4C targets or non-4C targets in MutuI (panel **c**) and Raji (panel **d**) cells. **f** Average expression levels for 4C target genes with EBNA1 binding sites (yellow, gene number: 124) or without binding sites (blue, gene number: 241) compared to non-4C associated genes (gray, gene number: 26,847). **g** Differential expression of shEBNA1 relative to shCTRL in Raji cells for 4C target genes with EBNA1 binding sites or without EBNA1 binding sites. Distribution for genes relative to 4C peaks was calculated for a randomly selected 100 genes for each category, and repeated 100 times. Boxplot showed the distribution (n = 100) of these averages. All boxplots show center quartiles, midlines show medians and whiskers extend to the data points, which are no more than 1.5× the interquartile range from the box. P-value were calculated by two-sided Mann–Whitney U test. **h**, **i** ChIP-qPCR for H3K9me3 or control IgG in shEBNA1 or shCTRL transduced Raji cells at EBNA1 binding sites in 4C target gene loci (panel **h**) or at non-4C gene loc with or without EBNA1 binding site loci (panel **i**). Error bar represents the standard deviation and P-value were calculated by two-sided Student t-test. n = 3. **j** Schematic model shows EBNA1 mediated genomic association between EBV episome and host chromosomes at locations enriched for H3K9me3, AT rich sequences, EBF1 and RBP-jK binding sites where are often observed in repressed neuronal genes. *P < 0.05; **P < 0.01.

## Discussion

EBV episome tethering to metaphase chromosomes is essential for viral genome maintenance in proliferating cells. While several mechanisms for metaphase attachment have been described, it remains unknown whether these mechanisms specify a location or function at the cellular chromosome attachment site. Here, we show that EBV genomes in BL cells attach to host chromosomes at locations enriched for EBNA1, EBF1, and RBP-jK binding sites, with elevated levels of H3K9me3 and flanked by AT-rich DNA (Fig. 6j). We also found that EBV attachments sites tend to occur at cellular genes that are transcriptionally silent and can be de-repressed by loss of EBNA1. These findings were limited to EBV latency associated with BL cell lines, and may be different in other latency cell types, especially LCLs, where attachment sites appear different than in BLs.

Early studies demonstrated that EBV attaches to the host chromosome through interactions involving EBNA1 and OriP. OriP was found to be the site of attachment to the nuclear matrix[51] and EBNA1 was found to localize to metaphase chromosomes[23,52]. Multiple mechanisms for EBNA1 attachment to metaphase chromosomes have been described, including interaction with AT-rich DNA[27], G-quadruplex RNA-binding[29], and interaction with host chromosome binding proteins, such as EBP2[26,31], HMGB2[22], nucleolin[32], and RCC1[33]. These metaphase attachment mechanisms are conferred, in part, through EBNA1 amino-terminal domains rich in arginine-glycine amino acids. However, EBNA1 has also been found to bind specific sites in the human genome through its sequence-specific DNA binding domain[19,20]. Whether these host chromosome EBNA1 binding sites contribute to EBV genome attachment was not known. Here, we have found that EBV genome attachment sites in Raji and MutuI BL cells frequently overlap with high-affinity EBNA1 binding sites in the host chromosome. The role of EBNA1 sequence-specific DNA binding as a docking mechanism at both the viral and host chromosome has not been previously considered (Fig. 6j). The ability of EBNA1 to interact homotypically through amino-terminal linking domains[36,38,39,53] and Zn-hook interactions[34] provides a plausible mechanism for EBNA1 to function as a bivalent sequence-specific tether for viral-host chromosome attachment.

Several other factors, in addition to EBNA1 site-specific host chromosome binding, correlated well with 4C attachment sites in BL cells. These include the colocalization of transcription factors EBF1 and RBP-jK, and enrichment of H3K9me3 and AT-rich DNA in flanking sequences. The AT-rich DNA is consistent with mechanisms proposed for EBNA1 interaction through its AT-hooks with AT-rich DNA stretches[27]. EBF1 and RBP-jK have been implicated in multiple aspects of lymphoid gene regulation and chromosome biology and are well-known sequence-specific

docking partners for EBNA2[10,54]. However, EBNA2 is not expressed in most BL cells. Therefore, it is more likely that EBF1 and RBP-jK binding sites on the viral genome may also mediate some form of homotypic interactions with host chromosome binding sites for EBF1 and RBP-jK, similar to that proposed for EBNA1. Interestingly, high affinity EBF1 and RBP-jK bindings sites around EBV OriP have been identified by ChIP-seq for both MutuI and Mutu-LCL[10]. The enrichment of H3K9me3, which is typically associated with heterochromatin and transcriptional silencing, suggests that EBV genomes are tethered to regions of the host genome that suppress transcription, especially in BL cells. We also observed that many of the silenced genes had neuronal functions and were enriched for protein kinase A pathway, potentially suggesting that EBV tethers at sites that are rarely expressed in B-lymphocytes or need to be silenced during latency. This is consistent with the restricted viral gene expression associated with type I latency typically observed in BL cells. It is also possible that EBV genomes are coordinately regulated with cellular genes in these chromosomal locations, perhaps through increasing the local concentration of shared transcription and epigenetic regulatory factors.

Multiple mechanisms of viral episome attachment have been reported for other DNA viruses. For some HPV strains, the viral genomes tether through an interaction of viral E2 with cellular BRD4 at fragile sites in the host chromosome[55] while HPV type 8 E2 tethers at ribosomal DNA sites[56]. One common theme is that HPV episomes are juxtaposed to host chromosomes at sites undergoing replication stress[57]. A similar variety of mechanisms have been reported for KSHV LANA. LANA binds to the core histones H2A and H2B[58], as well as to several other candidate receptors in the host chromatin, such as DEK[59], BRD2[60], TIP60, and telomere repeat factors (TRFs)[61]. These observations suggest that episomal DNA viruses utilize multiple mechanisms for attaching to host chromosomes, and that some of these mechanisms may be preferentially utilized in different cell types and/or viral strains.

A recent study using Hi-C methods examined EBV interactions with host chromosome in various cell types[48]. In this study, EBV was found to associate with gene poor, AT-rich locations restricted to a subset of chromosomes through an EBNA1 independent, but OriP-dependent mechanism[48]. There are several technical differences in this Hi-C study compared to ours. We utilized a higher-resolution 4C method and focused analysis primarily on BL cells Raji and MutuI rather than LCLs. Our attempts at using 4C in LCL resulted in lower resolution but was found to be correlated with published Hi-C studies in LCLs[47]. Our analysis suggests that episome tethering in LCLs may occur at more euchromatic sites than what we observed in BL cells. This

would be consistent with the more permissive EBV latency gene expression observed in LCL cells relative to BL cells. Another related study used capture Hi-C for analysis of KSHV DNA interactions during latency and reactivation[62]. This study did not address the host-chromosome target sites of the viral episome in latently infected cells. However, high-resolution 4C analysis did reveal extensive conformational structure during latency and lytic gene activation, and remodeling of these structures during this transition from latency to lytic.

In conclusion, we have used 4C methods to map common, high frequency interaction sites between the EBV and host genomes. We have found that these attachment sites in MutuI and Raji BL cells correlate with strong EBNA1 sequence-specific DNA binding sites, representing a new mode of viral genome tethering to the host chromosome. We further identified accessory factors, including EBF1 and RBP-jK, enrichment of H3K9me3 and AT-rich flanking DNA, as signature elements of EBV attachment sites. Attachment sites tended to overlap with transcriptionally repressed genes that could be derepressed by the depletion of EBNA1. These high-frequency attachment sites were not observed in LCLs, suggesting that different mechanisms of host tethering may predominate in different cell types. Our findings also suggest that virus-host genome interactions contribute to the epigenetic regulation of both viral and host gene expression and viral latency.

## Methods

**Cells**. MutuI is an EBV-positive Burkitt's Lymphoma cell (BL) line with type I latency (gift of J. Sample, Penn State HERSHEY Medical School) and Raji is an EBV-positive BL cell line with irregular type III latency (obtained from the ATCC, Catalog #CCL-86). The lymphoblastoid cell line (LCL) is an EBV-immortalized B-lymphoblastoid cell line with the Mutu virus strain, also referred to as Mutu-LCL. All B-cell-derived cell-lines were grown in RPMI medium with 12.5% fetal bovine serum (FBS) and antibiotics penicillin and streptomycin (50 U ml$^{-1}$). HEK293T cells (obtained from ATCC CRL-3216) were cultured in Dulbecco's modified Eagle's medium (DMEM) with 12.5% fetal bovine serum (FBS), 2 mM L-glutamine and antibiotics penicillin and streptomycin (50 U ml$^{-1}$).

**Virus production**. Lentiviruses were produced by cotransfecting the pLKO.1 shRNA expression plasmid packaging vectors pMD2.G and pSPAX2[63]. Cells in suspension were infected with lentiviruses carrying pLKO.1-puro vectors by spin infection at $450 \times g$ for 90 min at room temperature. The cell pellets were resuspended and incubated in fresh RPMI medium and then treated with 2.0 μg ml$^{-1}$ puromycin for 48 h after infection.

**4C-seq**. The 4C-seq was performed as previously reported with modifications (Fig. 1a)[64,65]. Briefly, five million cells were fixed in 1% paraformaldehyde for 10 min at 37 °C. Nuclei were permeabilized by incubation with 0.5% SDS at 62 °C for 10 min. DNA was digested with 100 units of MboI and ligated in the nucleus followed by in situ Hi-C protocol[47]. After reversal of crosslinks, DNA was digested with 100 units of Csp6I and re-ligated. For each primer viewpoint, a total 10 to 100 ng DNA was amplified by PCR. All samples were sequenced with Illumina NextSeq 500 High 75 bp single read. 4C-seq experiments from all viewpoints were carried out in biological replicates for all three strains; MutuI, Raji, and Mutu-LCL.

**4C-seq data analysis**. The 75 bp sequence single-end reads were grouped by cutadapt (version 1.11) based on the sequence from 4C bait location. Reads were aligned to human genome (hg19) using Bowtie2 (version 2.2.9) with iterative alignment strategy. Reads with low mapping quality (MapQ < 10) and reads that mapped to human repeat sequences were removed. Total aligned reads for each i-th position of non-overlapping 10 kb window ($N_i$) were calculated. Then, converted to the P-values using the Poisson formula:

$$Pi = 1 - \sum_{j=0}^{Ni} \lambda e^{-\lambda}/j! \qquad (1)$$

where $\lambda$ is equal to the average of reads for each 10 kb window (except EBV aligned reads). The significant peaks were defined using subcommand bdgpeakcall of MACS2 software (version 2.1.1) with parameters: at least P-value < $10^{-5}$ (option -c 5), minimum length of 20 kb (option -l 20000) and maximum gap of 10 kb (option -g 10000). Total significant peak number for each 10 kb were counted using 6 samples from same cells (DS/Cp-1/MboI/Bio1, DS/Cp-1/MboI/Bio2, DS/Cp-1/Csp6I/Bio1, DS/Cp-1/Csp6I/Bio2, DS/Cp-2/Csp6I/Bio1, and DS/Cp-2/Csp6I/Bio2). The 10 kb windows with 5 or 6 significant peaks out of 6 samples were defined as 'Common 4C peaks' for MutuI or Raji.

For the 4C data from Mutu-LCL, total aligned reads for non-overlapped 10 kb windows were calculated using surrounding 500 kb window and defined significant peaks using same method to MutuI and Raji. Total significant peak numbers for each 10 kb were counted using 10 samples from Mutu-LCL (FR-L/MboI/Bio1, FR-L/MboI/Bio2, FR-L/Csp6I/Bio1, FR-L/Csp6I/Bio2, DS/Cp-1/MboI/Bio1, DS/Cp-1/MboI/Bio2, DS/Cp-1/Csp6I/Bio1, DS/Cp-1/Csp6I/Bio2, DS/Cp-2/Csp6I/Bio1, and DS/Cp-2/Csp6I/Bio2). The 10 kb windows with 8 or more than 8 significant peaks out of 10 samples were defined as 'Common 4C peaks' for Mutu-LCL.

**EBV-human contacts from Hi-C data**. The Hi-C set from GM12878[47] was analyzed to extract the association between EBV episome and human chromosomes. Reference sequence from human genome (hg19) and EBV genome (V01555.2) were merged. Using the merged reference sequences, Hi-C data were processed as described previously[66]. Reads associated with the EBV genome were extracted and total reads for non-overlapping 10 kb windows of the human genome were calculated. P-value calculation and peak calling were performed same as 4C-seq data as described.

**Correlation analyses**. Correlation analyses were performed for 10 kb windows. The 10 kb windows were selected if (1) at least 3 hits out of 6 experiments, shown in Supplementary Fig. 8a, b, and (2) the maximum score of IgG in 10 kb window is not over 10. High IgG containing regions were filtered. The average scores for each 10 kb windows were calculated and sorted from highest score to lowest score order ($n = 14,167$ for MutuI, 22,718 for Raji).

Top 22,000 windows for Raji and top 14,000 for MutuI were grouped into 1000 (each group has 22 and 14, respectively). From the middle of 10 kb window, average ChIP-seq scores of upstream 500 kb and downstream 500 kb of each group were calculated for every 5 kb bin and plotted as heatmap. (Figs. 3a, 4a).

Top 22,710 windows for Raji and 14,160 windows for MutuI defined in above section were categorized into 10 groups (each group has 2271 and 1416, respectively). Average scores for each group were calculated for upstream/downstream 500 kb areas and plotted as a line graph (Figs. 3b, 4b).

Entire genomic regions were divided into non-overlapping 10 kb windows and calculated the average ChIP-seq enrichment. Average scores of randomly selected 100 windows from 4C targets ($n = 1569$ (Raji), 877 (MutuI)) were calculated. This process was repeated 100 times and the distribution was compared to the non-4C background. The same calculation was performed for the background. Average scores of randomly selected 100 windows from non-4C targets ($n = 307,997$ (Raji), 308,689 (MutuI)) were calculated. This process was repeated 100 times and distribution was used for the non-4C background. The average log2 ratio of 4C targets/background and p-value from two-sided t-test were reported (Supplementary Fig. 6).

**Circos plot**. Significant EBV-cellular 4C associations were plotted as circos graph format using the package 'circular' of R (version 0.3.3).

**ChIP-seq**. $5 \times 10^7$ MutuI or Mutu-LCL cells were crosslinked and sonicated to achieve a DNA fragment length of ~100–500 bp. Protein A/G Dynabeads were pre-incubated with 10 μg of rabbit anti-H3K9me3 (CiteAb cat# 39161) or rabbit anti-CTCF (EMD Millipore, cat# 07-729), and then incubated overnight with ChIP lysates. Dynabeads were then washed with ChIP-seq wash buffer (50 mM HEPES, pH 7.5, 500 mM LiCl, 1 mM EDTA, 1% NP-40, 0.7% Na-Deoxycholate, 1× protease inhibitors) for 5 times, then washed once with 50 mM NaCl in TE buffer. Bound DNA was eluted with ChIP-seq elution buffer (50 mM Tris-HCl, pH 8, 10 mM EDTA, 1% SDS), reverse-crosslinked at 65 °C, treated with RNase A (0.2 mg ml$^{-1}$) and proteinase K (0.2 mg ml$^{-1}$), extracted with phenol and chloroform, and subjected to qPCR validation. Validated ChIP DNA was isolated by agarose gel purification, ligated to primers, and then subject to Illumina based sequencing using manufacturer's recommendations (Illumina). ChIP-qPCR with ChIP DNA was performed in triplicate for each primer pair provided in Supplemental Table 7b.

**RNA-seq**. RNA was isolated using Trizol (Invitrogen) following manufacture instructions. Libraries were prepared from the extracted RNA using the QuantSeq 3'mRNA-seq Library Prep Kit-FWD (cat #15, Lexogen, Vienna, Austria) following the manufacture instructions using 1 μg of RNA per library, and then subject to Illumina based sequencing using manufacturer's recommendations (Illumina).

**ChIP-seq data processing**. Reads were aligned to the human genome (hg19) using Bowtie2 (version 2.2.9) with default option. Reads aligned to the same location were counted as only 1 with remaining reads were discarded by Picard (version 2.7.1) because those are likely derived from PCR artifact. Reads with low mapping quality (MapQ < 10) and from repetitive sequences were discarded. Pileup scores were calculated using Homer software (version 4.10.1)[67]. ChIP-seq scores correspond to the total number of mapped tags after normalizing to 10 million reads. Input DNA for ChIP was used for peak calling. ChIP-seq peaks were defined by HOMER software using the option -center -style factor -F 1 -P 0.0001 -fdr 0.05 (=fold change > 1, P-value < 0.0001 and FDR < 0.05) for EBNA1, EBF1, RBP-jk, CTCF, EBNA2, EBNA3 and PolII, while the alternative option -center -style histone -F 2 -P 0.0001 -fdr 0.05 (= fold change > 2, P-value < 0.0001 and FDR < 0.05) was used for other proteins.

**RNA-expression data processing**. The RNA-expression data for MutuI cells was generated in a previous study (GSE75385)[10] and reanalyzed for this study to calculate the average normalized scores of mRNA expression. RNA-seq data for Raji cells after transduction with shEBNA1 or shControl lentivirus was generated for this study using methods described previously[20]. RNA-seq reads were aligned to the human genome (hg19) using the STAR program (version 2.5.2). Reads assigned to exons were estimated by the RSEM program (version 1.2.31) and normalized using DESeq2 package (version 3.7).

**Tissue specific enrichment**. Genotype Tissue Expression (GTEx) data was downloaded from GTEx portal [https://www.gtexportal.org,[68]]. For each gene $Z$-scores were calculated using expression values across different tissues. Gene with a $Z$-score value for a tissue of at least 2 was considered as specifically expressed in the tissue. Only genes with known Entrez ID were considered and significance of enrichment was tested using Fisher Exact Test. $P$-values were adjusted for multiple testing using Benjamini-Hochberg correction and enrichments with the adjusted $p$-value < 0.05 were considered significant.

**Fluorescence in situ hybridization (FISH) microscopy**. Mitotic cells were collected and swelled in 0.075 M KCl for 25 min at 37 °C. Cell were fixed with 3 resuspensions of a methanol: acetic acid, 3:1 mix. Mitotic cells were dropped onto chilled clean glass slides and dried at least 4 h prior to staining. After drying, the cells were fixed for 5 min with 1% paraformaldehyde in PBS. Cells were washed and then dehydrated with ethanol wash series. Cells were co-stained with EBV probe (labeled with Oregon Green 488, ThermoFisher cat# A6374) and BACMID probe (CH17 BAC clones labeled with Alexa Fluor 594, Invitrogen cat# F32951) or 17p subtelomere probe (Cytocell cat# LPT17pG or LPT17pR) listed in Supplementary Table 4 in Hybridization Buffer B (Cytocell) overnight at 37 °C. Slides were washed two times with Wash1 (50% Formamide, 10 mM Tris pH7–7.5, 0.1% BSA) and additional two times with Wash2 (0.1 M Tris pH7–7.5, 0.15 M NaCl, 0.08%Tween), and then stained with DAPI. Slides were dehydrated with another ethanol series and mounted with Slow Anti Fade Gold mounting solution (Invitrogen). Mounted Slides were imaged on a Zeiss Axioimager Z1 fluorescence microscope with a 63x lens.

**Western blotting**. Cells were collected in RIPA buffer (50 mM Tris-HCL, pH6.7, 1% NP-40, 0.25% sodium deoxycholate, 150 mM NaCl, 1 mM EDTA) supplemented with 1 mM PMSF and 1× protease inhibitor cocktail (Roche). The anti-EBNA1 antibody, which was generated in rabbits immunized with bacterially expressed 1/500 dilution of EBNA1ΔGA (Pocono Rabbit Farms) and then affinity purified with the same antigen, and 1/100,000 dilution of anti-β-Actin Antibody (Sigma cat# A3854) were used for immunoblotting.

**RT-qPCR**. RNA was isolated using Trizol (Invitrogen) following manufacturer's instructions. Five microgram of RNA was DNase-treated and converted to random primed cDNA using Superscript IV kit (Invitrogen). cDNA was used as a template in subsequent PCR with gene-specific primers (Power SYBR green PCR master mix, Applied Biosystems). RT-PCR samples were analyzed by the 7900HT fast real-time PCR system (Life Technologies). Primer sequences are provided in Supplementary Table 7.

**Reporting summary**. Further information on research design is available in the Nature Research Reporting Summary linked to this article.

## Data availability

All 4C sequencing data has been deposited into the Gene Expression Omnibus (GEO) under accession code GSE129703. Sequencing data for RNA-seq after shEBNA1 depletion and ChIP-seq data are under accession code GSE129703 as well. The source data underlying Figs. 2e, f, h, 4e, 5a, and 6a–c, h, i and Supplementary Figs. 1a, d–f, 3b, 5a, 6a–c, 7a–f, 8a, b, f, and 9a, b are provided as a Source Data file.

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

## Acknowledgements

We thank the Andreas Wiedmer for excellent technical support. We acknowledge the support of the Wistar Cancer Center Genomics and Bioinformatics Core facilities. This work was supported by the National Institutes of Health (RO1 DE17336, RO1CA093606, and P30 CA010815) to P.M.L., the National Institutes of Health/National Institute of General Medical Sciences (R01GM124195) and National Institutes of Health/National Institute on Aging (P01AG031862) to K.N., and the National Research Foundation of Korea (NRF) grant funded by the Korea government (MSIT) (No. 2019R1F1A1061826 and 2019R1A4A1024764) to K.K.

## Author contributions

Conception and experimental design: K.K. and P.M.L. Methodology and data acquisition: K.K., A.D.L., O.V. and F.L. Computational analysis: H.T., A.K., L.C.S. and K.N. Interpretation of data: K.K., H.T., A.D.L., K.N. and P.M.L. Paper writing: K.K. and P.M.L.

## Competing interests

P.M.L. is a founder and advisor to Vironika, LLC; licensed technology to Cullinan-Apollo, Inc.; and served as advisor for GSK. The remaining authors declare no competing interests.
