## [Peer Review File · Nature Communications]

Reviewers' Comments:

Reviewer #1:

Remarks to the Author:

This study demonstrates tethering sites of EBV episomes on host chromosomes at high resolution for the first time. The authors selected several EBV genomic regions as baits to identify genomic regions where EBV episomes reside. The authors pre-selected the sites of genomic region (baits) based on previous observations and putative functions of EBNA1 and CTCF binding sites, those of which are known to interact with host chromosomes or function to gather genomic fragments to form genomic loops; these are well-justified approaches, based upon the author's previous publications. The results demonstrated EBNA1 but not CTCF binding sites colocalized with EBV episome binding sites. The results are very convincing and appropriate statistical analyses were included throughout the study. Based on bioinformatics approaches, the authors showed that EBV episomes more frequently reside where host chromatin is compacted. Accordingly, EBV episomes were found to be neighboring to genes associated with neuronal function in B cells; this is also partly due to physical association with RBP-Jk. Surprisingly, the authors showed that releasing EBV episomes from the host chromatin led to de-repression of silenced genes at the residential areas, indicating viral episomes themselves may play a role in regulation of histone modification at docking sites, perhaps by recruiting histone modifying enzyme complexes loaded on viral genes. In addition, by comparing two different latency types, the authors demonstrated that local histone modifications of the attached regions are associated with outcomes of types of EBV latency. Active EBV episomes are more frequently seen at the sites that have active histone marks. Overall, these are very exciting studies that have implications for not only gamma-herpesvirology but also for general epigenetic gene regulatory fields.

Followings are specific comments;

Methods;

1. Overall, the authors technical and computational analyses are sound. One minor question that this reviewer has is in regards to removal of PCR duplicates during the analysis of the 4C-seq data. Although it is mentioned in the methods that "repeat derived reads were removed", it is not clear if this refers to PCR duplicates. Secondly, it is not clear how valid di-tags are selected. Along these lines, it would be helpful to include a supplementary table that provides NGS and analysis metrics for each sample that includes number of total reads, reads passing filter, uniquely mapping reads, valid/invalid di-tags, etc.
2. Correlation analyses: This is nicely explained in the Methods. However, it would be helpful if there was additional explanation for the values that the "ChIP-Seq scores" refer to. Additionally, it would be helpful if slightly more description were added to the legend for Figure 1c, to better explain the Pearson correlation analyses (e.g., of ChIP-Seq scores).
3. It is very interesting why correlation of 4C peaks of DS/Cp-1 with LMP1/2 is less than with Qp even though LMP1/2 coding region is more closely located to DS/Cp regions. Does this mean EBNA1 binding predominantly determines EBV episomes docking site despite of local histone modification as your group determined before (Dheekollu J et al., J. Virol 2016). Please include some discussion.

The idea that genes near the genomic sites on human chromosomes where EBV genomes reside during latency are de-repressed following EBNA1 knockdown is interesting but not very convincingly demonstrated;

1. The authors show that EBNA1 KD, and presumably decreased EBV episome tethering, leads to up-regulation of selected EBV 4C-associated genes, as well as more globally by examining the expression

of the top 10% 4C target genes compared to those of non-4C targeting genes. While the latter addresses the aspect of a negative control, this reviewer wonders if KD of EBNA1 relieved repression of non 4C-enriched regions while still retaining EBNA1 binding at the regions interrogated.

2. How do the authors know that the EBNA1 depletion targets those EBNA molecules involved in tethering genomes at the locus examined? Since there is an excess of EBNA binding sites/cell identified by EBNA1 ChIP-seq (~1000/cell; these authors) over EBV genomes/cell (~50-100/cell), the KD could easily be targeting EBNA1 at other non-involved sites. Moreover, it would be helpful for reader if the authors could explain better why an initially cellular repressed B cell locus (i.e. neuronal genes) needs to be further repressed by EBV while residing in this niche. Isn't cellular repression sufficient enough for EBV genomes to hide out during latency? Why would further repression be needed? A combination of the author's DNA FISH with EBNA KD would strengthen the de-repression argument. For example, experiments with KD of EBNA1, probing the locus of interest with target-594 (red) and EBV (green), KD should give target (red) and lack of EBV (green). Finding a certain percentage of cells like this would add visual support to the claim of de-repression. In addition, combination of 3C studies with selected host genomic loci and its association with H3K9me3 marks with ChIP-qPCR (including other EBNA1 binding sites but not 4C sites) would strengthen the authors model.

3. The de-repression reported (Fig. 6B) is relative to shRNA control, as routinely done. However, in this case this reviewer thinks including unperturbed cells would be important to establish true de-repression. The use of shRNA as the control may stack outcomes in a positive direction. For example, a slight 2-fold decrease in expression induced by shRNA control, but no change in expression via the targeted shRNA will be scored as a 2-fold increase, yet there is actually no true effect relative to the basal level expression of untreated cells.

Minor;

Fig. 2C (merged) both arrow heads in the cell at the top of the figure point to non-overlapping (red) FISH signals.

According to the different overlap of 4C sites with H3K9me3, H3K27ac, EBNA1, EBF1 (these proteins show the highest ChIP-Seq scores on 4C sites, suppl Fig 5), it seems to be that EBV tethering sites are content-dependent. The authors might define the 4C sites into several classes and evaluate the gene signatures in each class separately. This reviewer considers that the proposed model is a little too simple. In minimum, the authors should discuss other possibilities.

Reviewer #2:

Remarks to the Author:

Epstein-Barr virus (EBV) episomes stably persist in replicating cells in part by EBNA1-mediated tethering to cellular chromosomes. This study uses circular chromosome conformation capture (4C) analysis to map the sites on the cellular chromosomes in Burkitt's lymphoma cells that are associated with EBV episomes. The analysis revealed an association with repressed chromatin and binding sites for EBNA1, EBF1 and RBP-jk and EBNA1 silencing was shown to lead to re-repression of these genes. The approach and results are novel and for the most part experiments are well done and well presented. However, a few points need to be addressed prior to publication as detailed below. In particular it is important to determine whether EBV episomes are released from the identified tethering sites after EBNA1 depletion.

Specific comments:

1. The study looks at 4 different regions of the EBV genome; FR, DS/Cp, Qp and an LMP1/2 locus. Most of the data analysis was done using overlapping peaks of DS/Cp-1 and DS/Cp-2 datasets. Since tethering of EBV genomes to chromosomes (and associated segregation function) is well characterized to occur through the FR element, it is not clear to me why they chose to do the detailed analysis on the DS/Cp data sets as opposed to the FR dataset.

2. In Fig 6, the authors silence EBNA1 and examine effects on transcription of genes associated with EBV tethering sites. This assumes that EBNA1 silencing will result in release of the EBV episomes from the chromosomal sites, however this has not been demonstrated using their system. It is critical to re-examine the association of the EBV episomes with the identified specific chromosomal attachment sites after depleting EBNA1, in order to determine whether all or subsets of these tethering sites are dependent on EBNA1. In addition to providing an important control, this analysis would also determine whether EBV attachment to predicted EBNA1 binding sites in the chromosomes is more dependent on EBNA1 than attachment to other sites chromosome sites.

3. In Fig 6b, the authors show that EBNA1 depletion increases transcription of genes associated with EBV episomes to varying degrees. Some of these effects are quite small and may not be statistically significant. Effects of EBNA1 depletion on other genes not associated with EBV episomes are not shown, raising the possibility that EBNA1 depletion could have a general effect on transcription. The authors need to show results with several genes that are not EBV-associated in order to demonstrate that the effect is specific to EBV-associated genes.

4. There are several places where important references are missing:

a. Pg 3, paragraph 2, statement on the similar structures of the EBNA1 and LANA DBDs. References are missing for the LANA structure.

b. Pg3, paragraph 3, the statement on mapping the metaphase chromosome binding domains of EBNA1 is missing all of the references from the Frappier lab who did a large body of work on this topic. References are Shire et al 1999 PMID:10074103, Wu et al 2000 PMID:11265753, Wu et al 2002 PMID:11836426, Shire et al 2006 PMID:1669900, Nayyar et al 2009 PMID:19887584.

c. In the same paragraph there is reference to the linking domains of EBNA1 that mediate homotypic interactions. These were actually called either looping or linking domains by different groups and references to the early work on this topic are missing. They are Frappier and O'Donnell 1991 PMID:1660154, Goldsmith et al 1993 PMID:8388506, Laine and Frappier 1995 PMID:8537346.

Reviewer #3:

Remarks to the Author:

In the manuscript NCOMMS-19-08963, Kim et al characterized the chromatin interactions between EBV episomes and the host genome in Burkitt lymphoma (BL) cells, using the 4C technology. The authors found that EBV preferentially associate with transcriptionally silenced genes with enrichment in neuronal function in BL cells. These EBV-interacting cellular genomic regions are enriched in EBNA1, EBF1 and RBP-jK binding sites as well as the repressive histone mark H3K9me3, and surrounded by AT-rich sequences, which is consistent with previous findings in the literature. Moreover, depletion of EBNA1 from EBV latently infected BL cells led to a transcriptional de-repression of these silenced genes.

Specific concerns

1. Similar work has been published last year (Moquin et al, 2018, J Virology, Volume 92 Issue 3

e01413-17).

2. what are the copy number of EBV in MutuI, Raji, Mutu-LCL, and GM12878 cells, respectively? How signals were normalized to the copy number of HBV? Is it possible that the differences in peaks between MutuI and Raji is due to differences in copy number?

3. How big is the EBV genome size? What is the sequence diversity between individual EBV genomes?

3. It's odd that the number of overlap sites between the common 4C peaks and the EBV integration sites in Raji is so low (Supplementary Fig. 2c), as each EBV integration site should exhibit much stronger 4C signal than signals between episomal EBV and host genome (which are equivalent to inter-chromosomal interactions). This probably is a copy number issue.

4. it is strange/interesting to see that EBV-interacting peaks are mostly enriched for repressive marks in MutuI and Raji, but active marks in GM12878 cells (Fig 4e and Fig S5). The authors may offer more detailed explanation.

Reviewers' comments:

Reviewer #1 (Remarks to the Author):

This study demonstrates tethering sites of EBV episomes on host chromosomes at high resolution for the first time....Overall, these are very exciting studies that have implications for not only gamma-herpes virology but also for general epigenetic gene regulatory fields.

Response: We appreciate the supportive comments.

Followings are specific comments;

Methods;

1. Overall, the authors technical and computational analyses are sound. One minor question that this reviewer has is in regards to removal of PCR duplicates during the analysis of the 4C-seq data. Although it is mentioned in the methods that “repeat derived reads were removed”, it is not clear if this refers to PCR duplicates.

Response: In the 4C method, all sequenced reads of 4C-seq library were started from view point primers near Mbol or Csp6I restriction sites (Fig.1a, Methods/4C-seq) differently with the reads of Hi-C data which is generated from random sonication sites. Therefore, it is not appropriate to remove sequencing duplicates in our analysis of the 4C-seq data, since we cannot discriminate unique sequenced reads from PCR duplicates. Alternatively, we converted all scores into P-values and selected very reproducible peaks among 6 (for Raji and Mutul) or 8 (for Mutu-LCL) samples to avoid the overestimation of PCR duplicates.

Secondly, it is not clear how valid di-tags are selected.

Response: We did not use paired-ends, so there are no di-tags in this 4C method. We have sequenced reads as 75bp single-end, instead of paired-end reads. Valid 4C-seq reads were selected by valid 4C-seq adaptors. Then we have removed reads with low mapping quality (MapQ < 10) and repeat driven reads. Read numbers of each filtering step were summarized in Supplementary Table S2 and S3.

Along these lines, it would be helpful to include a supplementary table that provides NGS and analysis metrics for each sample that includes number of total reads, reads passing filter, uniquely mapping reads, valid/invalid di-tags, etc.

Response: We provide detailed metrics on the NGS data (Supplementary Figure S1, Supplementary Tables S1, S2, and S3) that include information on the numbers of intra (EBV-EBV) and inter (EBV-human) hybrid reads.

2. Correlation analyses: This is nicely explained in the Methods. However, it would be helpful if there was additional **explanation for the values that the “ChIP-Seq scores”**

refer to. Additionally, it would be helpful if slightly more description were added to the legend for Figure 1c, to **better explain the Pearson correlation analyses** (e.g., of ChIP-Seq scores).

Response: We now provide additional explanation:

ChIP Seq scores are now defined in the Methods sections on ChIP-seq data processing:

“ ChIP-seq scores correspond to the total number of mapped tags after normalizing to 10 million reads.”

We updated Fig.1C as shown below:

“ (C) Heatmap showing the Pearson's correlation coefficients of 4C-seq scores for 10kb non-overlapped windows between baits from same (top) or different strains (bottom).”

Minor

According to the different overlap of 4C sites with H3K9me3, H3K27ac, EBNA1, EBF1 (these proteins show the highest ChIP-Seq scores on 4C sites, suppl Fig 5), it seems to be that **EBV tethering sites are content-dependent**. The authors might **define the 4C sites into several classes and evaluate the gene signatures in each class separately**. This reviewer considers that **the proposed model is a little too simple**. In minimum, the authors should discuss other possibilities.

Response: We further stratified the EBV tethering genes into two groups based on the presence of EBNA1, since our data indicates EBNA1 is one of key factor for EBV tethering (Fig.3 and Fig.6b). We provide new analyses in Fig 6f and g showing that EBNA1-associated 4C peaks are associated with genes that are more transcriptionally repressed and more likely to be de-repressed after shEBNA1 depletion. Additionally we provide new Supplementary Figure 4a showing that EBNA1 associated 4C sites have greater enrichment of RBP-jK, EBF1, and H3K9me3. We also extend Supplementary Figure 7 by analyzing IPA pathways for EBNA1 associated 4C peaks relative to all 4C and 4C lacking EBNA1 peaks.

3. It is very interesting why correlation of 4C peaks of DS/Cp-1 with LMP1/2 is less than with Qp even though LMP1/2 coding region is more closely located to DS/Cp regions. Does this mean EBNA1 binding predominantly determines EBV episomes docking site despite of local histone modification as your group determined before (Dheekollu J et al., J. Virol 2016). Please include some discussion.

Response: We did not perform detailed analysis of Qp or LMP1/2 interactions as we focused on those most likely linked through OriP. We agree that a more extensive analysis of these other interactions may reveal some interesting findings, but we consider that beyond the scope of this manuscript.

The idea that genes near the genomic sites on human chromosomes where EBV genomes reside during latency are **de-repressed following EBNA1 knockdown is**

interesting but not very convincingly demonstrated;

1. The authors show that EBNA1 KD, and presumably decreased EBV episome tethering, leads to up-regulation of selected EBV 4C-associated genes, as well as more globally by examining the expression of the top 10% 4C target genes compared to those of non-4C targeting genes. While the latter addresses the aspect of a **negative control**, this reviewer wonders **if KD of EBNA1 relieved repression of non 4C-enriched regions** while still retaining EBNA1 binding at the regions interrogated.

Response: We provide new data (Fig. 6c) showing that KD of EBNA1 does not affect transcription of several non-4C associated genes with EBNA1 ChIP-Seq binding sites, nor genes that lack EBNA1 binding sites. We show this trend in a genome-wide analysis of all 4C vs non4C genes with or without EBNA1 sites by integrating RNA expression data with 4C and EBNA1 ChIP-Seq data (Fig. 6f). Furthermore, we show that KD of EBNA1 leads to a general derepression of EBNA1 associated 4C genes relative to all other genes based on new RNAseq data (Fig. 6g).

2. How do the authors know that the EBNA1 depletion targets those EBNA molecules involved in tethering genomes at the locus examined? Since there is an excess of EBNA binding sites/cell identified by EBNA1 ChIP-seq (~1000/cell; these authors) over EBV genomes/cell (~50-100/cell), the KD could easily be targeting EBNA1 at other non-involved sites.

Response: We agree that the EBNA1 depletion is problematic for the reasons mentioned, as well as for the primary role of EBNA1 in binding the EBV episomes. Nevertheless, the primary evidence supporting the model is the highly significant correlation between 4C interactions and EBNA1 ChIP-Seq sites. Our new analysis shows that genes associated with both 4C attachment and EBNA1 binding sites are more likely to be repressed relative to all other genes (Fig. 6f) and derepressed by EBNA1 KD (Fig. 6g). We also provide new FISH (Supplementary Figure 8c and d) and site-specific 3C data (Supplementary Figure 8e and f) showing that EBNA1 KD leads loss of specific associations between EBV and cellular genomes.

Moreover, it would be helpful for reader if the authors could explain **better why an initially cellular repressed B cell locus (i.e. neuronal genes) needs to be further repressed by EBV while residing in this niche**. Isn't cellular repression sufficient enough for EBV genomes to hide out during latency? Why would further repression be needed?

Response: In the Discussion, we provide a new comment to help the reader understand how EBV tethering may regulate cellular gene expression, including stabilizing repression of heterochromatic genes. How and why this additional repression occurs remains speculative, but within the framework of epigenetics and stochastic gene activation.

“It is also possible that EBV genomes are coordinately regulated with cellular genes in these chromosomal locations, perhaps through increasing the local concentration of shared transcription and epigenetic regulatory factors.”

A combination of the author’s DNA FISH with EBNA KD would strengthen the de-repression argument. For example, **experiments with KD of EBNA1, probing the locus of interest with target-594 (red) and EBV (green), KD should give target (red) and lack of EBV (green)**. Finding a certain percentage of cells like this would add visual support to the claim of de-repression.

Response: We provide new data in Supplemental Figure 8c and d showing a loss of EBV FISH signal after EBNA1 KD.

In addition, combination of 3C studies with selected host genomic loci and its association with H3K9me3 marks with ChIP-qPCR (including other EBNA1 binding sites but not 4C sites) would strengthen the authors model.

Response:(1) We provide new data in Figure 6h and I showing a loss of H3K9me3 after EBNA1 KD. (2) We provide new data in Supplemental Figure 8e and f that an EBV-cellular interaction as measured by 3C is reduced after EBNA1 KD.

3. The de-repression reported (**Fig. 6B**) is relative to shRNA control, as routinely done. However, in this case this reviewer thinks including unperturbed cells would be important to establish true de-repression. The use of shRNA as the control may stack outcomes in a positive direction. For example, a slight 2-fold decrease in expression induced by shRNA control, but no change in expression via the targeted shRNA will be scored as a 2-fold increase, yet there is actually no true effect relative to the basal level expression of untreated cells.

Response: Since there are many manipulations in the shRNA experiment, including lentivirus infection and puromycin selection for 5-9 days, we consider it essential to compare to a similarly treated control, and problematic to compare to an unperturbed cell.

Minor;

Fig. 2C (merged) both arrow heads in the cell at the top of the figure point to non-overlapping (red) FISH signals.

Response: Although it may appear as though there is no overlap, all of the arrowheads indicate signals that overlap in Z-stacks. Nevertheless, we have removed the one arrow at the top and right point that is objectionable.

According to the different overlap of 4C sites with H3K9me3, H3K27ac, EBNA1, EBF1 (these proteins show the highest ChIP-Seq scores on 4C sites, suppl Fig 5), it seems to be that **EBV tethering sites are content-dependent**. The authors might **define the 4C sites into several classes and evaluate the gene signatures in each class**

separately. This reviewer considers that **the proposed model is a little too simple.** In minimum, the authors should discuss other possibilities.

Response: We have added another level of stratification to separate classes of genes. Specifically, we have stratified by the presence or absence of an EBNA1 ChIP-Seq peak within the gene locus (Figure 6f and g, Supplementary Figures 4a, 7, 8a and b). We find that genes associated with 4C peaks that also have EBNA1 peaks are even further enriched in H3K9me3, EBF1, and RBP-jK. When we focused on the top 10% of 4C enriched genes, we found an enrichment of neuronal genes.

Reviewer #2 (Remarks to the Author):

Epstein-Barr virus (EBV) episomes stably persist in replicating cells in part by EBNA1-mediated tethering to cellular chromosomes. This study uses circular chromosome conformation capture (4C) analysis to map the sites on the cellular chromosomes in Burkitt's lymphoma cells that are associated with EBV episomes. The analysis revealed an association with repressed chromatin and binding sites for EBNA1, EBF1 and RBP-jK and EBNA1 silencing was shown to lead to re-repression of these genes. The approach and results are novel and for the most part experiments are well done and well presented. However, a few points need to be addressed prior to publication as detailed below. In particular it is important to determine **whether EBV episomes are released from the identified tethering sites after EBNA1 depletion.**

Response: We now show that EBNA1 depletion leads to a loss of EBV episomes by FISH (Supplementary Figure 8c and d) and by 3C (Supplementary Fig. 8e and f).

Specific comments:

1. The study looks at 4 different regions of the EBV genome; FR, DS/Cp, Qp and an LMP1/2 locus. Most of the data analysis was done using overlapping peaks of DS/Cp-1 and DS/Cp-2 datasets. Since tethering of EBV genomes to chromosomes (and associated segregation function) is well characterized to occur through the FR element, it is not clear to me **why they chose to do the detailed analysis on the DS/Cp data sets as opposed to the FR dataset.**

Response: We chose to focus on DS/Cp data set to identify highly significant EBV tethering sites. 'The Pearson's correlation coefficient was as high as 0.95 for DS/Cp-1 or 2 (Supplementary Fig. 1c, d). We also examined the reproducibility of datasets from two sides of a bait, Mbol or Csp6I. The Pearson's correlation coefficient was 0.66–0.77 for FR-L (Left side), 0.81–0.93 for DS/Cp-1, and 0.55–0.70 for LMP1/2 (Supplementary Fig. 1c, e). So we chose DS/Cp data sets since DS/Cp data sets gave us the highest correlation for all pair-wise comparisons (Fig. 1c, Supplementary Fig. 1d,e,f).

In the revised manuscript, we provide a new analysis that incorporates the overlapping 4C peaks between DS/Cp and FR-L (Supplementary Figure 4b). This analysis revealed

a greater enrichment of EBNA1 ChIP-Seq binding sites, further supporting the model that EBNA1 sequence-specific binding is a major determinant of EBV genome tethering to the host chromosome.

2. In Fig 6, the authors silence EBNA1 and examine effects on transcription of genes associated with EBV tethering sites. This assumes that EBNA1 silencing will result in release of the EBV episomes from the chromosomal sites, however this has not been demonstrated using their system. **It is critical to re-examine the association of the EBV episomes with the identified specific chromosomal attachment sites after depleting EBNA1**, in order to **determine whether all or subsets of these tethering sites are dependent on EBNA1**.

In addition to providing an important control, this analysis would also determine **whether EBV attachment to predicted EBNA1 binding sites in the chromosomes is more dependent on EBNA1 than attachment to other sites chromosome sites**.

Response: While we were unable to repeat the genome-wide 4C under conditions of shEBNA1 depletion, we do provide new data showing the EBNA1 tethering site at chr2:15.6Mb loses EBV interaction after EBNA1 depletion using 3C methods (Supplemental Figure 8e and f). We also show by FISH the bulk of EBV tethering sites are reduced after shEBNA1 depletion (Supplemental Figure c and d). We also show that genes overlapping 4C tethering sites and colocalizing with EBNA1 are more likely de-repressed by shEBNA1 (Figure 6g).

3. In Fig 6b, the authors show that EBNA1 depletion increases transcription of genes associated with EBV episomes to varying degrees. Some of these effects are quite small and may not be statistically significant. Effects of EBNA1 depletion on other genes not associated with EBV episomes are not shown, raising the possibility that EBNA1 depletion could have a general effect on transcription. The authors need to show results with **several genes that are not EBV-associated in order to demonstrate that the effect is specific to EBV-associated genes**.

Response: From response to Reviewer 1. We provide new data (Fig. 6c) showing that KD of EBNA1 does not affect transcription of several non-4C associated genes with EBNA1 ChIP-Seq binding sites, nor genes that lack EBNA1 binding sites. We show this trend in a genome-wide analysis of all 4C vs non4C genes with or without EBNA1 sites by integrating RNA expression data with 4C and EBNA1 ChIP-Seq data (Fig. 6f). Furthermore, we show that KD of EBNA1 leads to a general derepression of EBNA1 associated 4C genes relative to all other genes based on new RNAseq data (Fig. 6g).

4. There are several places where important references are missing:
a. Pg 3, paragraph 2, statement on the similar structures of the EBNA1 and LANA DBDs. References are missing for the LANA structure.

Response: Corrected.

b. Pg3, paragraph 3, the statement on mapping the metaphase chromosome binding domains of EBNA1 is missing all of the references from the Frappier lab who did a large body of work on this topic. References are Shire et al 1999 PMID:10074103, Wu et al 2000 PMID:11265753 , Wu et al 2002 PMID:11836426, Shire et al 2006 PMID:1669900, Nayyar et al 2009 PMID:19887584.

Response: These references have been added.

c. In the same paragraph there is reference to the linking domains of EBNA1 that mediate homotypic interactions. These were actually called either looping or linking domains by different groups and references to the early work on this topic are missing. They are Frappier and O'Donnell 1991 PMID:1660154, Goldsmith et al 1993 PMID:8388506, Laine and Frappier 1995 PMID:8537346.

Response: These references have been added.

Reviewer #3 (Remarks to the Author):

In the manuscript NCOMMS-19-08963, Kim et al characterized the chromatin interactions between EBV episomes and the host genome in Burkitt lymphoma (BL) cells, using the 4C technology. The authors found that EBV preferentially associate with transcriptionally silenced genes with enrichment in neuronal function in BL cells. These EBV-interacting cellular genomic regions are enriched in EBNA1, EBF1 and RBP-jK binding sites as well as the repressive histone mark H3K9me3, and surrounded by AT-rich sequences, which is consistent with previous findings in the literature. Moreover, depletion of EBNA1 from EBV latently infected BL cells led to a transcriptional depression of these silenced genes.

Specific concerns

1. Similar work has been published last year (Moquin et al, 2018, J Virology, Volume 92 Issue 3 e01413-17).

Response: There are many differences between our work and Moquin et al. We reference and discuss Moquin et al. We identified numerous EBV 4C peaks through chromosome with 10 kb resolution. It is much higher resolution than previous study (Moquin et al used 1000 kb bin to identify single nearest genes). As a result, we identify more specific genetic loci for EBV tethering. We used 4C-seq analysis to focus on EBV tethering sites, but previous work extract EBV tethering sites from whole genome association acquired from Hi-C. Moreover, we validated our genomics data by using FISH and 3C methods.

2. what are the copy number of EBV in Mutul, Raji, Mutu-LCL, and GM12878 cells, respectively? How signals were normalized to the copy number of HBV? **Is it possible that the differences in peaks between Mutul and Raji is due to differences in copy**

number?

Response: We determined the total genomic copy number by qPCR and standard curve method for all cell lines used in this study (Supplementary Figure 5d).

The number and strength of peaks could be affected by copy number of EBV. LCL strains (Mutu-LCL and GM12878) have lower episome copy number than BLs (Mutu and Raji). It is possible that copy number correlates to a different mechanism of attachment, and we discuss this possibility since we find differences in tethering sites between BL and LCLs.

3. How big is the EBV genome size? What is the sequence diversity between individual EBV genomes?

Response: The EBV genome for Mutu is 171,687 bp. There is some sequence diversity (polymorphisms and deletions) between EBV genomes. It is possible that some of these variations may contribute to different tethering preferences, but all competent episomes must have an intact EBNA1 and OriP.

3. It's odd that the number of overlap sites between the common 4C peaks and the EBV integration sites in Raji is so low (Supplementary Fig. 2c), as each EBV integration site should exhibit much stronger 4C signal than signals between episomal EBV and host genome (which are equivalent to inter-chromosomal interactions). This probably is a copy number issue.

Response: The reviewer raises an interesting question as to whether episome tethering is related to aberrant integration. Our analysis suggests that there is very little overlap between tethering and integration sites, suggesting that these two events are not related. Since the copy number of BLs is around 50-100, it is not clear how copy number may be related to the integration sites. We hypothesize that EBV integration may occur during lytic cycle, and likely to occur at other sites in the host chromosome that are distinct from tethering sites we have identified here.

4. It is strange/interesting to see that EBV-interacting peaks are mostly enriched for repressive marks in Mutu and Raji, but active marks in GM12878 cells (Fig 4e and Fig S5). The authors may offer more detailed explanation.

Response: As reviewer pointed out, EBV tethering sites on host chromosomes are substantially different between BLs (Mutu and Raji) and LCL (Mutu-LCL and GM12878). These differences may be due to the differences in host cell types or latency type. We have added more discussion to this point. A more detailed experimental analysis of these differences is beyond of scope for this manuscript.

Reviewers' Comments:

Reviewer #1:

Remarks to the Author:

The authors addressed most of our concerns. Several specifics that should be included in the manuscript for further clarification are listed below.

Please clarify.

Line 431. "P-values from a Poisson distribution of reads (except EBV aligned reads) were used for peak calling by MACS2 software (version 2.1.1) with the parameters: '-c 5 -l 20000 -g 10000'.

Based on the provided parameters, the authors should clarify how they generated the bedgraph files in more detail as this is not the typical use of MACS2. The authors should elaborate and mention that they used MACS2 bdgpeakcall. This reviewer understands that they are using MACS2 for the 4C data, which is significantly different from typical ChIP-seq. In such case, it will be important for the authors to provide some reasoning for the parameters used for the analyses. This reviewer thinks that those parameters make a difference in outcome of the results.

Line 199 and others. EBV non-4C target sites.

The description needs to be more specific. It is important to explain how the non-4C target sites are determined. Please provide information regarding how many non-4C target sites are used for analyses.

For the 4C and ChIP-seq enrichment heatmap (e.g. Fig 3A) and metagene (e.g. Fig 3b), 500 kb flanking the 4C peak center, i.e. genomic fragments with size of 1mb, were viewed. It is not clear if it is appropriate to look at ChIP-Seq signals at a 500 kb window, because TF binding signals are often very narrowly localized. A 500 kb window might contain many peaks for TF's and across several genes; this might favor the broader ChIP-signals such as H3K9me3. As this approach (comparing 4C data set with ChIP-seq) is relatively new, additional justification for the experimental setting will be helpful.

Please define how the 4C peaks are assigned to specific genes, especially for the 10% peaks mentioned in fig 5. The 70 sites seem to span across the genome with 100-400 kb.

Fig 6f. 6g. Please include specific numbers. How many genes are included in the EBNA binding genes, EBNA non-binding genes, or non 4C-targeting genes. Please describe how the authors selected the EBNA binding genes.

Figs. 6b/6c values are listed as relative expression. What are the mean Ct values used in the calculations going from a repressed to de-repressed state in Fig 6b? In the interest of helping both the reviewer and readers more fully evaluate the results presented, for Figs. 6b/6c please provide the mean Ct values for each mRNA analyzed here, shCTRL vs. shEBNA1.

ChIP-seq:

Although the authors provided some description in method, this reviewer still could not completely follow how the ChIP-seq scores were generated. The authors state that they "correspond to the total number of mapped tags after normalizing to 10 million reads." It is not clear if a baseline control (e.g., non-Chip-enriched input DNA) was utilized and what software was utilized downstream after normalization, such as Homer (as mentioned in Methods). Also, although "average scores" are referred to, it seems that replicates were not performed (i.e., based on page 3 of the "203049_1_additional_review_material_4042375_pwywd1.pdf" document). Along those lines, if

replicates were not performed for the ChIP-Seq experiments presented, it is suggested to perform at least duplicates according to "ChIP-seq guidelines and practices of the ENCODE and modENCODE consortia" (Landt et al. Genome Research. doi/10.1101/gr.136184.111). Finally, it would be very helpful for the ChIP-Seq data to be deposited at NCBI also.

There are some corrections needed for proper usage of English as follows: Correlation Analyses (line 455), Circos plot (lines 478-470), RNA-seq (lines 497-501), Chip-Seq data processing (lines 503-509) and RT-qPCR (lines 552-553).

Yoshihiro Izumiya

Reviewer #2:

Remarks to the Author:

My previous comments were largely addressed. However I noted a few minor issues that still need correction:

1. The sentence on line 283-284 refers to qPCR data showing a total drop in EBV copy number upon EBNA1 silencing. The FISH results are referenced in this sentence (suppl Fig 8C and d) but where is the qPCR data?
2. On line 283, the word "silencing" is missing from "As expected, EBNA1 led to...".
3. References 38 and 39 should be added to the statement on line 332 "...EBNA1 to interact homotypically through amino-terminal linking domains..."

Reviewer #3:

Remarks to the Author:

In the revised manuscript, the authors added results demonstrating that EBNA1 depletion lead to specific de-repression of EBNA1 repressed genes and concurrently the release of EBV episomes from the identified tethering sites. This reviewer is also satisfied with the authors' responses to the other concerns raised by the reviewers in the previous round.

Reviewers' comments:

Reviewer #1 (Remarks to the Author):

The authors addressed most of our concerns. Several specifics that should be included in the manuscript for further clarification are listed below.

Please clarify.

1. Line 431. "P-values from a Poisson distribution of reads (except EBV aligned reads) were used for peak calling by MACS2 software (version 2.1.1) with the parameters: '-c 5 -l 20000 -g 10000'.

Based on the provided parameters, the authors should clarify how they generated the bedgraph files in more detail as this is not the typical use of MACS2. The authors should elaborate and mention that they used MACS2 bdgpeakcall. This reviewer understands that they are using MACS2 for the 4C data, which is significantly different from typical ChIP-seq. In such case, it will be important for the authors to provide some reasoning for the parameters used for the analyses. This reviewer thinks that those parameters make a difference in outcome of the results.

Response: As suggested by the reviewer, we described in more detail how 4C-seq peaks were defined ("4C-seq data analysis") in Method section:

"Total aligned reads of each i -th position of non-overlapping 10 kb window (N_i) were calculated. Then, converted to the P-values using the Poisson formula:

$$P_i = 1 - \sum_{j=0}^{N_i} \lambda e^{-\lambda} / j!$$

where λ is equal to the average of reads for each 10 kb window (except EBV aligned reads). The significant peaks were defined using subcommand "bdgpeakcall" of MACS2 software (version 2.1.1) with parameters: at least P-value < 10^{-5} (option "-c 5"), minimum length of 20 kb (option "-l 20000") and maximum gap of 10 kb (option "-g 10000")."

2. Line 199 and others. EBV non-4C target sites.

The description needs to be more specific. It is important to explain how the non-4C target sites are determined. Please provide information regarding how many non-4C target sites are used for analyses.

Response: We now provide a more detailed explanation of how the non-4C target sites were selected in Method/ Correlation analysis (underlined is newly added text).

" (iii) **Enrichment of ChIP-seq on EBV tethering sites.** Entire genomic regions were divided into non-overlapping 10 kb windows and calculated the average ChIP-seq enrichment. Average scores of randomly selected 100 windows from 4C targets ($n = 1569$)

(Raji), 877 (Mutul)) were calculated. This process was repeated 100 times and the distribution was compared to the non-4C background. The same calculation was performed for the background. Average scores of randomly selected 100 windows from non-4C targets (n = 307,997 (Raji), 308,689 (Mutul)) were calculated. This process was repeated 100 times and distribution was used for non-4C background. The average log₂ ratio of 4C targets/ background and p-value from two-sided t-test were reported. (Supplementary Fig. 6). “

3. For the 4C and ChIP-seq enrichment heatmap (e.g. Fig 3A) and metagene (e.g. Fig 3b), 500 kb flanking the 4C peak center, i.e. genomic fragments with size of 1mb, were viewed. It is not clear if it is appropriate to look at ChIP-Seq signals at a 500 kb window, because TF binding signals are often very narrowly localized. A 500 kb window might contain many peaks for TF's and across several genes; this might favor the broader ChIP-signals such as H3K9me3. As this approach (comparing 4C data set with ChIP-seq) is relatively new, additional justification for the experimental setting will be helpful.

Response: As reviewer points out, 500 kb range might contain multiple TF binding signals. However, our data clearly indicates a strong correlation of 4C-seq peaks, with specific TF binding peaks, such as EBNA1, RBPjK, and EBF1. These all have very sharp ChIP-Seq peaks. In contrast, CTCF, EBNA2, and EBNA3 did not correlate well with 4C peaks. Additionally, EBNA1 binding peaks did not show strong correlation with the control non-4C binding sites using a similar 500 kb window. Furthermore, the correlation analyses with Hi-C related experiments tend to use wide range of chromosome since Hi-C related peaks are broadly enriched on chromosome (see Hou, C.,, & Corces, V. G., 2012, *Molecular Cell* 2012 ;48(3):471-84. doi: 10.1016/j.molcel.2012.08.031)

4. Please define how the 4C peaks are assigned to specific genes, especially for the 10% peaks mentioned in fig 5. The 70 sites seem to span across the genome with 100-400 kb.

Response: Any genes which are included in the 10% peaks are selected and examined for the tissue specific enrichment. The defining of the 10% peaks is as follows: We selected the top 10% of 4C peaks based on width since Hi-C related peaks are broadly enriched on chromosome as mentioned above. We found 70 sites that overlapped with these peaks in both Raji and Mutul cells (Fig. 5a, b). We demonstrated that these 70 regions are positively correlated with the average and maximum strength of reads in both cell lines (Fig. 5c) and broadly distributed across multiple chromosomes (Fig. 5d). It is described in the main text.

5. Fig 6f. 6g. Please include specific numbers. How many genes are included in the EBNA binding genes, EBNA non-binding genes, or non 4C-targeting genes. Please describe how the authors selected the EBNA binding genes.

Response: The genes overlapped with significant EBNA ChIP-seq peaks were defined as EBNA binding genes. The numbers of genes are now described in the revised legend for Figure 5::

“f) Average expression levels for 4C target genes with EBNA1 binding sites (yellow, gene number: 124) or without binding sites (blue, gene number: 241) compared to non-4C associated genes (grey: gene number 26,847). g) Differential expression of shEBNA1 relative to shCTRL in Raji cells for 4C target genes with EBNA1 binding sites or without EBNA1 binding sites. Distribution for non-targeted genes was calculated for a randomly selected 100 genes and repeated 100 times. Boxplot showed the distribution (n=100) of these averages.”

6. Figs. 6b/6c values are listed as relative expression. What are the mean Ct values used in the calculations going from a repressed to de-repressed state in Fig 6b? In the interest of helping both the reviewer and readers more fully evaluate the results presented, for Figs. 6b/6c please provide the mean Ct values for each mRNA analyzed here, shCTRL vs. shEBNA1.

Response: We provide for the reviewer the average Ct value of RT-qPCR shown in Fig 6b,C, and now include this as an additional supplementary Figure 9.

7. Although the authors provided some description in method, this reviewer still could not

completely follow how the ChIP-seq scores were generated. The authors state that they “correspond to the total number of mapped tags after normalizing to 10 million reads.” It is not clear if a baseline control (e.g., non-Chip-enriched input DNA) was utilized and what software was utilized downstream after normalization, such as Homer (as mentioned in Methods). Also, although “average scores” are referred to, it seems that replicates were not performed (i.e., based on page 3 of the “203049_1_additional_review_material_4042375_pwywd1.pdf” document). Along those lines, if replicates were not performed for the ChIP-Seq experiments presented, it is suggested to perform at least duplicates according to “ChIP-seq guidelines and practices of the ENCODE and modENCODE consortia” (Landt et al. Genome Research. doi/10.1101/gr.136184.111). Finally, it would be very helpful for the ChIP-Seq data to be deposited at NCBI also.

Response: We provide additional information on the ChIP-Seq methods in this manuscript. Biological replicates were not performed for ChIP-seq experiments. ChIP-seq data for unpublished H3K9me3 and CTCF were deposited to GEO with accession number GSE129703. Accession number of other proteins were described previously and are included in Supplementary Table 5.

“Input DNA in ChIP were used for peak calling. ChIP-seq peaks were defined by HOMER software using the option “-center -style factor -F 1 -P 0.0001 -fdr 0.05” (= fold change > 1, P-value < 0.0001 and FDR < 0.05) for EBNA1, EBF1, RBP-jk, CTCF, EBNA2, EBNA3 and Pol2, while the alternative option “-center -style histone -F 2 -P 0.0001 -fdr 0.05” (= fold change > 2, P-value < 0.0001 and FDR < 0.05) was used for other proteins.

8. There are some corrections needed for proper usage of English as follows: Correlation Analyses (line 455), Circos plot (lines 478-470), RNA-seq (lines 497-501), Chip-Seq data processing (lines 503-509) and RT-qPCR (lines 552-553).

Response: Thank you for finding these errors. We have carefully read over and corrected errors in proper usage of English.

Reviewer #2 (Remarks to the Author):

My previous comments were largely addressed. However I noted a few minor issues that still need correction:

1. The sentence on line 283-284 refers to qPCR data showing a total drop in EBV copy number upon EBNA1 silencing. The FISH results are referenced in this sentence (suppl Fig 8C and d) but where is the qPCR data?

Response: For shRNA depletion studies, we monitored EBV copy number by FISH (Supplemental Figure 8C and D). The revised text now indicates that copy number was measured by FISH only.

2. On line 283, the word “silencing” is missing from “As expected, EBNA1 led to...”.

Response: corrected.

3. References 38 and 39 should be added to the statement on line 332 “...EBNA1 to interact homotypically through amino-terminal linking domains...”

Response: References have been added.

Reviewer #3 (Remarks to the Author):

In the revised manuscript, the authors added results demonstrating that EBNA1 depletion lead to specific de-repression of EBNA1 repressed genes and concurrently the release of EBV episomes from the identified tethering sites. This reviewer is also satisfied with the authors' responses to the other concerns raised by the reviewers in the previous round.

Response: Thank you for your review.

Reviewers' Comments:

Reviewer #1:

Remarks to the Author:

The authors addressed all concerns.